# Bi-Triggering Energy Harvesters: Is It Possible to Generate Energy in a Solar Panel under Any Conditions?

Krzysztof A. Bogdanowicz

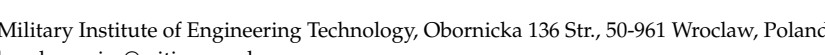

Military Institute of Engineering Technology, Obornicka 136 Str., 50-961 Wroclaw, Poland;
bogdanowicz@witi.wroc.pl

**Abstract:** In this review, the concept of a hybrid solar cell system, called all-weather solar cells, a new view on energy harvesting device design, is introduced and described in detail. Additionally, some critical economical, technological, and ecological aspects are discussed. Due to drastic global climate changes, traditional energy harvesting devices relying only on solar energy are becoming less adaptive, hence the need for redesigning photovoltaic systems. In this work, alternative energy harvesting technologies, such as piezoelectric and triboelectric devices, and photoelectron storage, that can be used widely as supporting systems to traditional photovoltaic systems are analysed in detail, based on the available literature. Finally, some examples of all-weather solar cells composed of dye-sensitized solar cells (DSSC) and silicon solar cells, often modified with graphene oxide or phosphors materials, as new perspective trends in nanotechnology are presented. Two types of solar cell triggers are analysed: (i) solar cells working during day and night (DSSC with phosphors materials), and (ii) solar cells working under sun and rain conditions (piezoelectric and triboelectric silicon or DSSC solar cells).

**Keywords:** all-weather solar cells; energy harvesting; rain energy harvesting; piezoelectric effect; triboelectric effect; DSSC; silicon solar cells

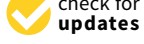



## 1. Introduction

Both the advancement in technology and the rise of living standards has resulted in increased energy usage. Decentralised energy systems are the reasonable response to what is needed in the current situation. The self-powered feature in electronic applications is likely to be developed in the form of ambient energy harvesting. An increasing interest in renewable energy aligns with the constant growth of the global population and concerns about environmental degradation issues and encourages the development of non-pollutant energy sources, one of which is the harvesting of solar energy. Current challenges for solar harvesting systems are mainly cost reduction and the increase of sunlight-to-energy conversion efficiency [1–3]. Ceca et al. [4] highlight in their work, that one of the important factors, such as geographical location, daily cloud coverage, temperature, precipitation, and total light exposure, need to be considered in order to achieve high energy conversion.

To meet the increasing demand for electric energy, a concept of "all-weather solar cell" has been proposed by professor Qunwei Tang [5,6]. Tang and co-workers designed an energy harvester that generates energy from the sun and rain by adding an extra layer on top of the solar panel which generates energy from the rain drops. For the first time, the idea of harvesting raindrop energy was presented by Guigon et al. [7] with the purpose of providing energy for small devices that consume less power. It can now be provided from energy scavenged from sources like mechanical vibrations, acoustic signals, and solar energy. Since then, some researchers have investigated the possibility of harvesting energy from rain, which is the most frequent occurring phenomenon on the globe above the Tropic of Cancer and Capricorn. The example of such devices includes both electrostatic and piezoelectric energy harvesting [8], or piezoelectric/triboelectric and electromagnetic

devices [9]. Another approach proposed by Tang [6,10] is all-weather solar cells, harvesting solar energy from a wider spectral range during the daytime and during dark periods using the incorporated long persistent phosphors (LPP) which are fluorescent-emitting and absorb a specific light wavelength during sunny hours and then emits it during dark hours.

This review article focuses on summarizing the different methods that can be implemented in bi-triggering solar cell systems in order to increase overall energy generation with specific emphasis on:

- Piezoelectric devices for rain energy harvesting;
- Triboelectric devices for rain energy harvesting;
- Photo-electron storage properties of LLP (long persistent phosphors) materials;
- Concepts of all-weather solar cells presented as is schematically shown in Figure 1.

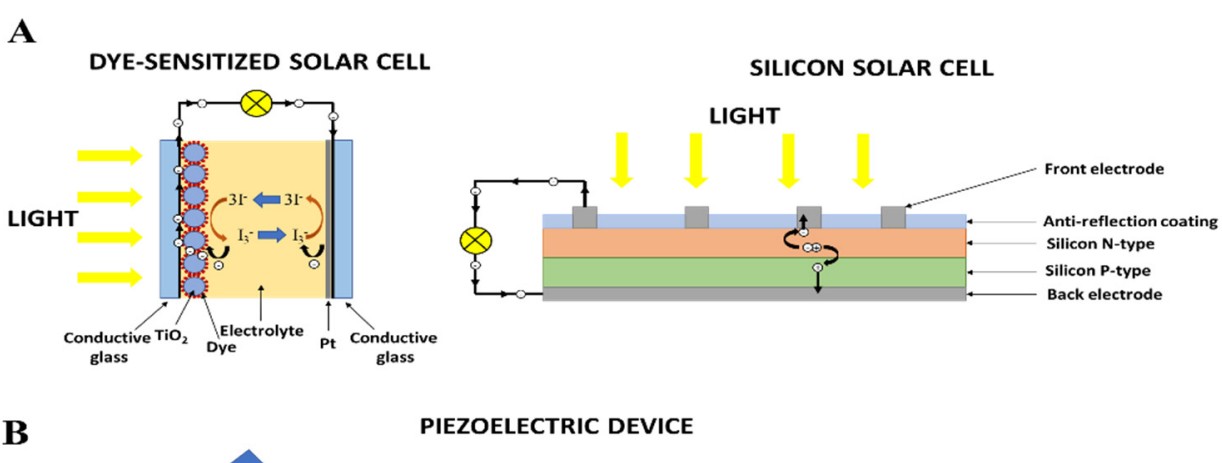

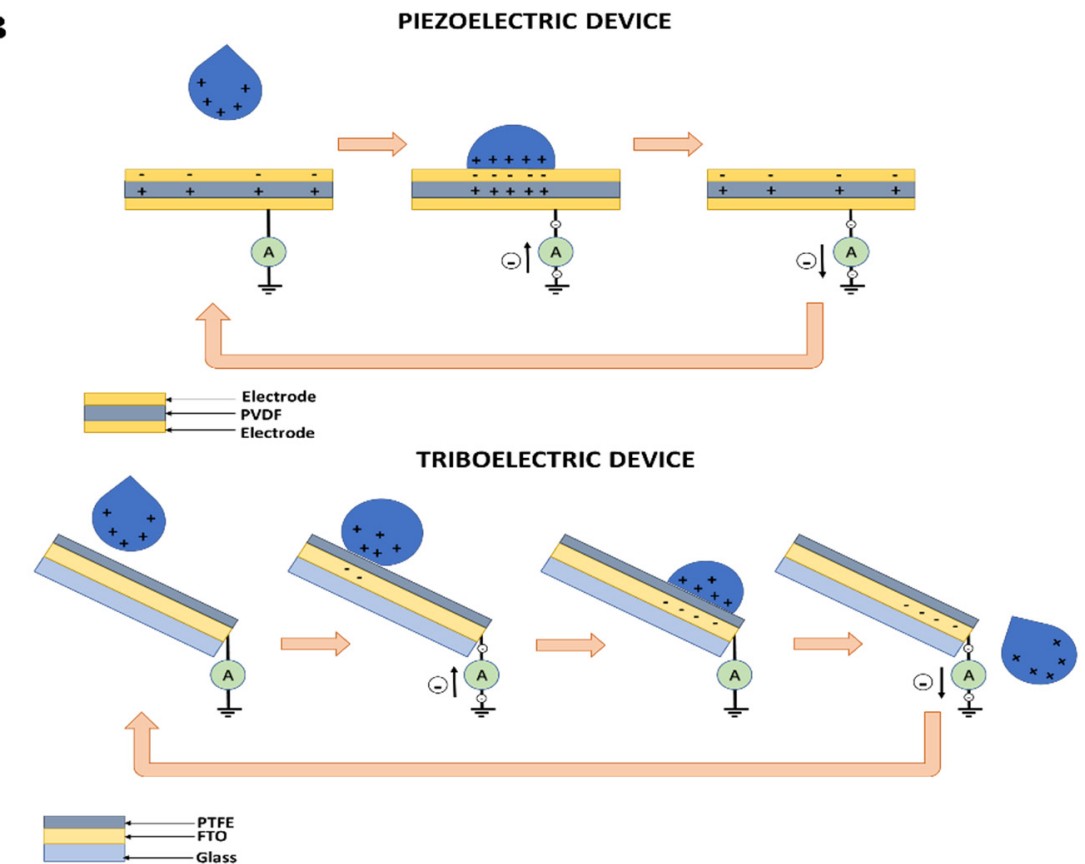

**Figure 1.** *Cont.*

**PHOTO-ELECTRON STORAGE PROPERTIES OF LLP MATERIALS**

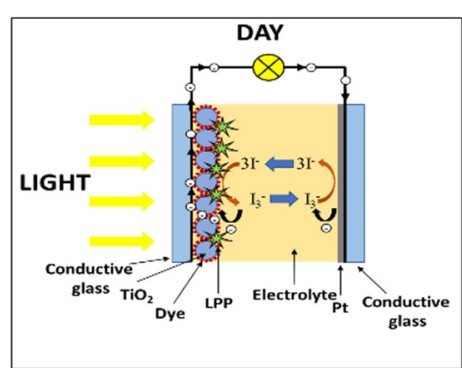 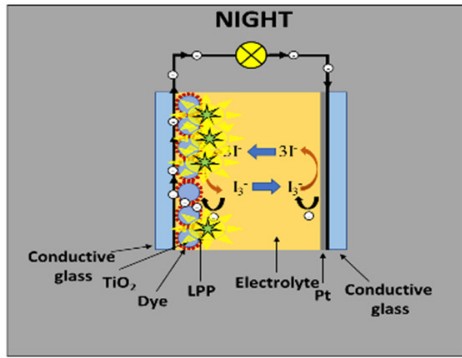

**ALL-WEATHER SOLAR CELL**

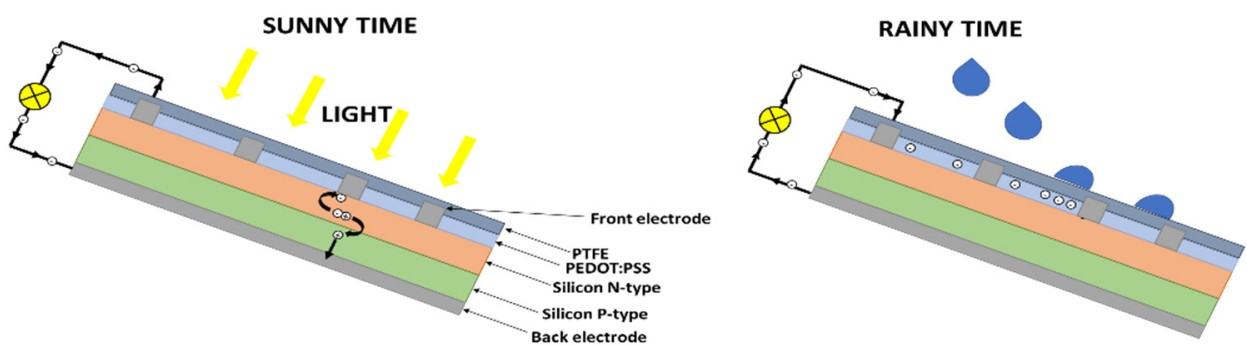

**Figure 1.** The principle of operation of DSSC and silicon solar cells (**A**) and graphical representation of above mentioned co-harvesting systems (**B**).

Two types of solar cell triggering are analysed: (i) solar cells working during the day and night (DSSC with phosphors materials), and (ii) solar cells working under sun and rain conditions (piezoelectric and triboelectric silicon or DSSC solar cells).

Finally, the answer for the titled question can be found in this review based on the limited amount of scientific papers, but which present all the important aspects of the analysed subjects. One of the goals of this review is to present possible diverse approaches of energy harvesters that could be combined in a single device to produce electric energy from several independent sources. To the best of my knowledge, this is the first review paper about all-weather solar cells.

## 2. Energy Harvested from Rain

### 2.1. Piezoelectric Devices

The piezoelectric phenomenon has been used to convert mechanical motion into electrical energy in an efficient way, as presented in Figure 2 for a piezoelectric device for a rain energy harvester.

## PIEZOELECTRIC DEVICE

**Figure 2.** Piezoelectric transducer for the mechanic to electric energy conversion of rain.

Due to their ability to convert mechanical energy into electrical energy, piezoelectric materials, both inorganic ones (lead zirconium titanate, barium titanate) [11,12] and organic ones (e.g., polymer ferroelectrics) [13,14], are the focal point of interest in the area of energy harvesting. These materials have become more vital because of the need for self-powered systems.

Perera et al. [15] described experiments in which three rain harvester types were analysed under three different rain intensity scenarios (light, moderate, and heavy). A maximum power of $2.231 \times 10^{-29}$ W was calculated theoretically for a 1 m$^2$ area for a polyvinylidene fluoride (PVDF) membrane under heavy rain conditions. Individual rain parameters are introduced and the method of calculating individual parameters is presented. Since the rain is usually characterized by its rate (the quantity of rain over a specified time period most often expressed in mm/h) and DSD (the number of drops that are measured as a function of their diameter in a given sample). To measure these two parameters, a laser precipitation monitor sensor was used. As far as raindrop impact was concerned, this force is generated as the drop falls onto the piezoelectric energy harvester surface. Furthermore, spreading, splashing, and bouncing are the behaviours that a drop exhibits on impact with a surface. The most dominant of which is splashing, which has been reported to reduce the amount of energy generated. With regard to raindrop accumulation, a thin layer of water forming on the piezoelectric beam surface could cause a resistive force impacting the dynamic properties of the structure and, in turn, decrease the value of voltage produced by the system as it is described by Wong et al. [16].

In the literature, piezoelectric compounds are characterised by higher piezoelectric coefficients and brittleness, while the latter are flexible, can withstand greater strains, but have lower piezoelectric coefficients [7,15,17–19]. Bhavanasi and co-workers [17] selected poly(vinylidene fluoride-co-trifluoroethylene) (PVDF-TrFE, see Figure 3) due to its piezoelectric coefficients. The improvement in the energy harvesting performance in devices using piezoelectric/ferroelectric materials in the form of one-dimensional nanostructures were created, including hybrid devices i.e., those made by combining different vibration energy harvesting mechanisms in one device. One of the materials attracting significant interest in this area is graphene oxide (GO). The numerous functionalities are attributed to the diversity of possible functional groups, resulting in local negative charge domains in

the GO layered structure. The work [17] described bilayer film formation via drop casting forming graphene oxide: a ferroelectric PVDF-TrFE-layered structure. This device preparation resulted in the improvement of the energy harvesting performance of ferroelectric PVDF-TrFE films. Further, by combining the piezoelectric (ferroelectric) PVDF-TrFE with a GO as the charged material, it is possible to harvest both piezoelectric and electrostatic energy due to applied force; the GO's high Young's modulus and dielectric constant number in the efficient transfer of mechanical to electrical energy. Used in this work, GO exhibited capacitance of approximately $1.7 \times 10^{-4}$ C cm$^{-3}$ for an Al/p-Si/SiO$_2$/GO/Au capacitor.

## P[VDF-TrFE] copolymer

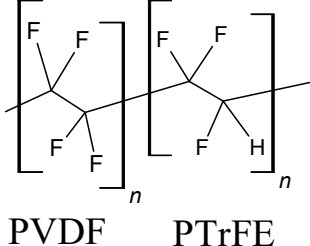

PVDF       PTrFE

**Figure 3.** General structure of poly(vinylidene fluoride-co-trifluoroethylene).

Experimental methods described in this work [17] involved ITO as the bottom electrode and gold electrode sputtered on to the PVDF-TrFE films. The results obtained with regard to the poled PVDF-TrFE films showed a maximum voltage output of ca. 2.2 V under 0.32 MPa at 1 Hz of dynamic compression pressure, as compared with bilayer films with the poled PVDF-TrFE layer which showed a maximum voltage output of ca. 4.3 V under the same dynamic compression pressure. All three samples are shown in Figure 4.

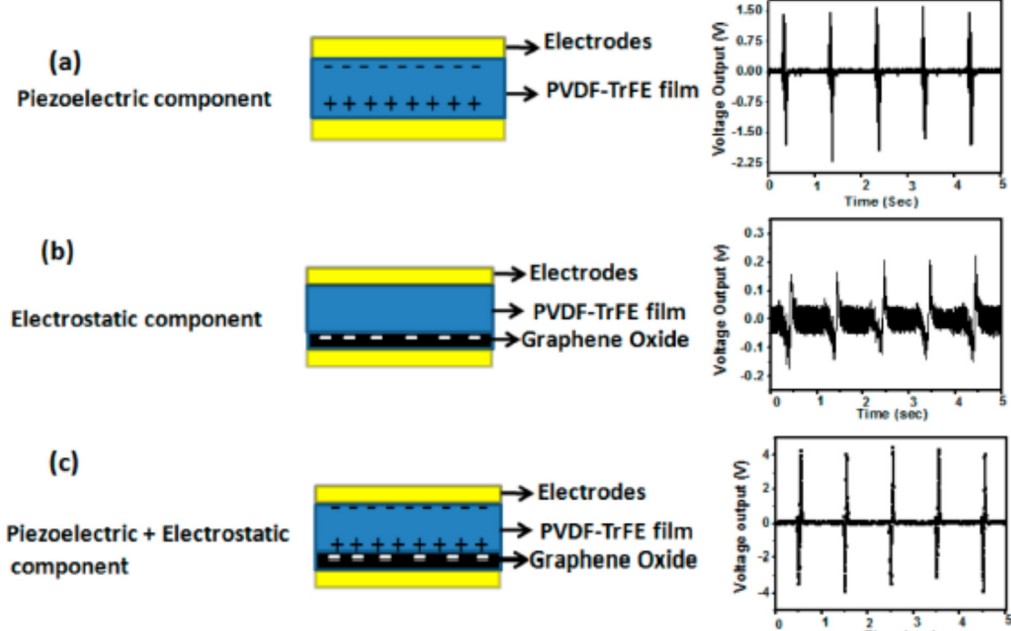

**Figure 4.** Working mechanism of the bilayer films energy harvesting device: (**a**) piezoelectric voltage output from the poled PVDF-TrFE film; (**b**) electrostatic component (voltage output from the bilayer films, with the nonpoled PVDF-TrFE layer); (**c**) voltage output from the bilayer films device consisting of both electrostatic and piezoelectric components. Reproduced with permission from [17] Copyright 2016 American Chemical Society.

The device was made as bilayer films consisting of poled PVDF-TrFE (thickness 32 μm) and GO (approximate thickness: 19 μm) with the needle-shaped ferroelectric β phase. As far as graphene oxide morphology was concerned, a corrugated surface can be observed, and the cross-sectional view reveals perfect adhesion of the PVDF-TrFE and GO. The device was flexible, bendable, and rollable, which makes it easily integrable with flexible electronics. In order to optimize the device performance, GO layer thickness in bilayer films was equalled 3 μm (with peak output voltage obtained ca. 3 V), 20 (whose peak output voltage approximated 4 V), and 30 μm, the peak output voltage remained at the level of 4 V due to graphene oxide, which screened the electrical output. Regarding the power output, in the case of PVDF-TrFE and the bilayer, the voltage was measured for resistors with the internal resistances in a range between 43 kΩ and 6.7 MΩ, with maximum power value being greater for bilayer films 4.41 μW cm$^{-2}$, while this value was 1.17 μW cm$^{-2}$ for the PVDF-TrFE. This value was measured for a load resistor, giving the information for real-time application, when compared to measurements obtained in open circuit conditions (voltage) and short circuit conditions (current). What is interesting is the almost linear output increase with increasing compression pressure. The device manufactured withstood 1000 cycles with no degradation.

Wong et al. [18] focuses on the mechanical vibration as an energy source using piezo-electrics (i.e., lead zirconatetitanate-PZT or polyvinylidene fluoride-PVDF) which can be called "smart materials" capable of converting vibrations into electricity direct by piezo-electric effect. They study the possibility of harvesting the energy impact of a raindrop with a piezoelectric beam and focus on four key parameters determining the influence of rain, which are rain rate, raindrop count, rainfall depth, and drop site distribution (DSD), on the energy harvesting device. Three rain events were recorded and studied by the authors. The first was rain rate and its correlation with instantaneous voltage. The study case was a rain event: the first lasting 250 min, max. rain rate of 381.9 mm/h produced 511.53 μW; the second lasting 240 min, max. rain rate of 344.9 mm/h gave 1441 μW; the third one during 301 min, max. rain rate of 407.3 mm/h produced 3.85 mW. Raindrop count was plotted vs. instantaneous energy; however, their correlation is not strong. Raindrop count was measured via laser precipitation monitor sensor, whereas the instantaneous energy graph was computed by integrating the instantaneous power for a given period. The variation of possible energy output is also dependent on the location of the impact point due to the beam deflection, impacting the clamped end of the piezoelectric beam. Rainfall depth and energy accumulation graphs indicate the total energy equaled 155.6 mJ, coming from the first rain event, and the total accumulated rainfall depth −84.2 mm. For the second these values were 438.9 μJ and 172.9 mm, respectively, while the third event produced the largest values of 2076 μJ and 124.0 mm, respectively. To estimate the volume and size of water on the piezoelectric beam surface, and to enable the modal water mass calculation, the mass coefficient was added. The maximum values obtained from the three rain events were the following: 0.30, 0.33, and 0.34, respectively. The values justifiably fluctuate depending on the number of drops impacting the surface. It was found that the rain duration had no influence on the total energy harvested; however, rain rate is a vital factor: the higher the rain rate, the more power generated. The voltage obtained for the piezoelectric beam depends on the size of the raindrop and the location where the impinging raindrop impacted. Impact force depends on the drop diameter. Additionally, they point out the disadvantages of energy harvesting from raindrop impact, namely its cost (it is more expensive than fossil fuel energy), the low power obtained when piezoelectric materials are used, and its dependence on atmospheric conditions such as rain and advise coupling such systems with other energy harvesters.

Ong et al. [19] proposed the modification of a lead zirconatetitanate (PZT) rain energy harvester prototype (bimorph cantilever beam) in order to improve its performance in both a spray-type simulator and during actual rain. The rain simulator with various nozzles was applied to generate rain drops with different drop size distributions and intensities. They developed a number of equations with regard to the motion, behaviour,

and factors impacting the cantilever beam. This included an irregular water film formation on the surface impacting its properties, including the damping coefficient, stiffness, and mode shape. Raindrops themselves were classified into five classes, specified in the original work [19] regarding the laser precipitation monitor, and the impact of the force exerted by the rain droplets onto the piezoelectric beam was studied by assessing the excitation force. To model the force, an analogy of a ball impacting a plate was used, mimicking the differences of drops splashing and spreading and not sticking onto the impact surface [7,15]. Studies found that bigger output was generated in the case where the water droplet impacted on the piezoelectric tip; however, during real rain individual droplets impacted the beam randomly. With improvement of the performance in mind, the authors introduced a mechanism of obtaining the droplets of the same size from drops of different sizes, reducing the possibility of low voltages generated by smaller droplets. The second scenario involved the redirection of the water droplets to the tip of the PZT beam, thus, increasing the impact force (see Figure 5).

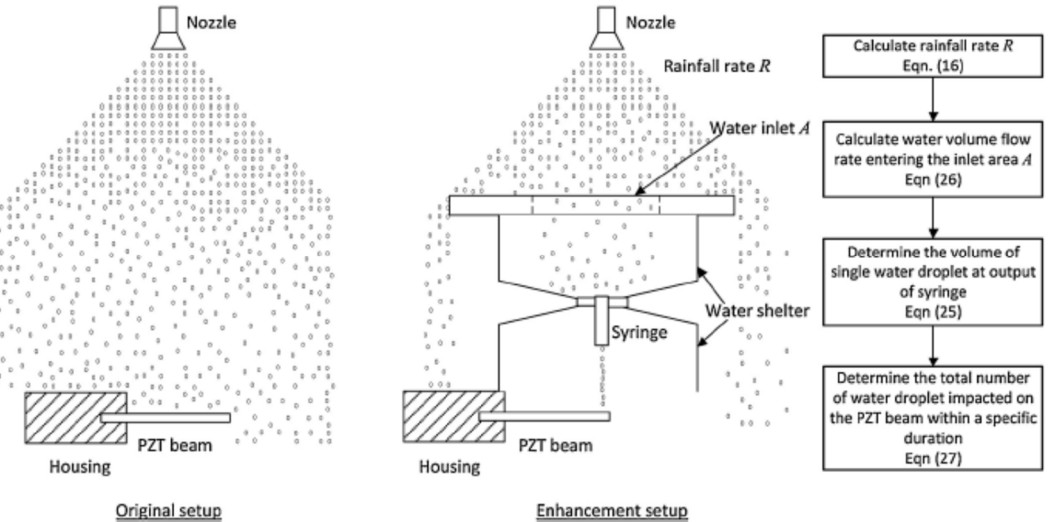

**Figure 5.** Schematic diagram of the original and enhancement setup. Reproduced with permission from [19] © 2021 Elsevier B.V.

Experiments in a rain simulator and in actual rain were conducted. The rain simulator simulated the rainfall rates from 32 mm/h to 99 mm/h. The raindrops were found to be approximately 0.5 mm in diameter, and the maximum measured was 5 mm. Larger diameter raindrops suffered from a higher resistance in the air, forcing their split into two droplets. For the rain precipitation at 99 mm/h, the basic setup (without additional adjustments) provided random peak voltages (due to variations in droplet size and random impact positions), with the average of an approximately 0.2 V voltage peak. For the enhanced setup, the average peak voltage equalled ca. 0.95 V, with the variation in voltage being minor. The average voltage peak for the simulator and experimental study was found to be 0.99 V and 0.95 V, respectively, due to some slight variations of the raindrop diameter and rainfall rate. When the theoretical calculation of the peak voltage was compared the results were very similar: 0.99 V vs. 0.91 V. Since the energy harvested had positive correlation to the rainfall rate, as more raindrops impacted on the harvester and more energy was harvested. The performance of the enhanced setup gave an improvement ranging from 208% to 490% of the total harvested energy. Both the sets were also tested under actual rain conditions. The rain lasted for 35 min and the maximum rain precipitation rate was found to equal 298.8 mm/h. For the original setup in this experiment, the maximum peak voltage was 7.428 V, much higher when compared to the results for the enhanced setup, chiefly due to the higher raindrop impact force. The rain harvester with original setup generated uneven values of the highest peak voltages, due to the randomness

of the raindrop impact on the PZT beam, whereas the enhanced setup registered consistent voltage generation. At the time values above 25 min, for the two configurations, it appeared to be saturated regarding the energy accumulation, as rainfall rate was low after the 25 min mark. Considering the above, energy generated at the duration after 25 min was lower comparing to the one generated prior, reaching a plateau state. The total energy accumulated during the rain for the original setup was 1695 µJ and for the enhanced setup equalled 1811µJ. It means a small improvement, despite the fact that the enhanced setup performed relatively worse in actual rain. However, higher peak voltage generated with the original setup did not ensure greater energy generation in total, compared to the enhanced setup, likely as a result of the random droplets present, yielding negligible voltage. The enhanced setup could potentially become a continuous power supply due to the consistent voltage which it provided.

In the summary of this part of the review, it can be mentioned that the best electrical properties were found by Wong et al. [18] for the piezoelectric device with the PVDF/PZT beam. In Table 1, the characteristic electrical parameters of these devices are presented.

**Table 1.** Summary of selected electrical properties for piezoelectric devices.

| Device Configuration | Voltage Output [V] | Energy Accumulated [µJ] | Power Density [µW cm$^{-2}$] | Ref. |
|---|---|---|---|---|
| PVDF membrane | - | - | $2.231 \times 10^{-19}$ [a] | [15] |
| Ag/PVDF-TrFE(10µm)/GO(20µm)/Ag | $4.00 \pm 0.23$ | - | 4.41 [b] | [17] |
| PVDF/PZT beam | 7.6 | 2076 | 213 | [18] |
| PZT beam | 7.428 | 1811 | - | [19] |

a—theoretical study, b—load 1 MΩ.

As has been shown, piezoelectric effect can be used to generate energy from rain in an effective way under various atmospheric conditions of weather.

*2.2. Triboelectric Devices*

Another technology for harvesting rain energy is by triboelectric transducer. The mechanism of energy generation is based on the charge transfer caused by moving water on hydrophobic polymer films covering electrically conducting electrodes, as presented in Figure 6. What is important is that the hydrophobic films are transparent, allowing their deposition directly over the solar cells, and thus, simultaneous energy both from the sun and rain can be harvested [20–22].

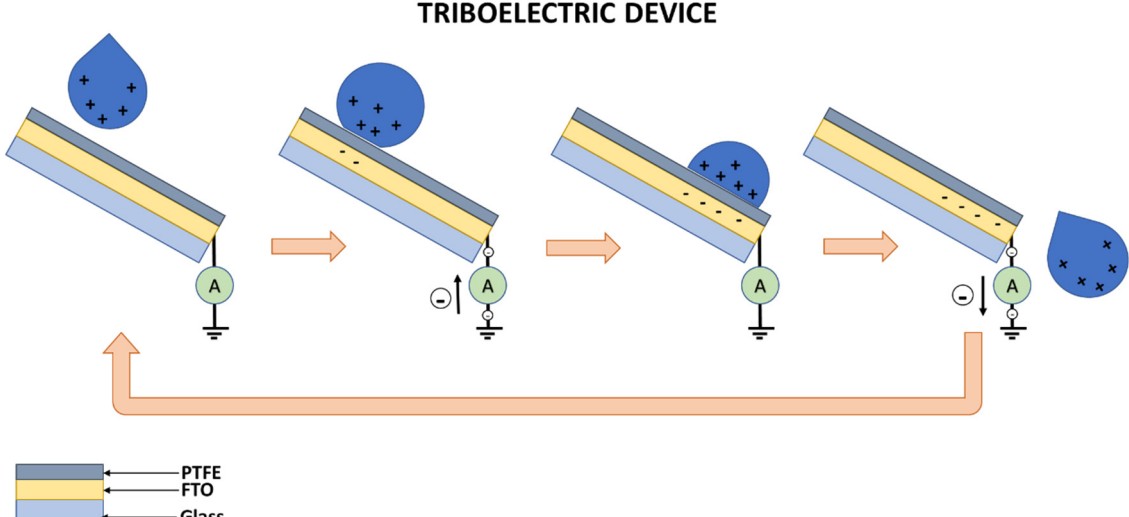

**Figure 6.** Triboelectric transducer for electrostatic to electric energy conversion of rain.

Liang and co-workers [23] proposed a transparent triboenergy nanogenerator (T-TENG) that uses a single electrode, preferable for harvesting mechanical energy from continuously flowing water, a thin (1 μm) film with an overlap of hydrophobic, less than 3 μm thick poly(tetrafluoroethane) (PTFE) film, was used. The transmittance of the constructed device was found to be larger compared to the glass substrate. It reached a 10 V output peak-to-peak open-circuit and 2 μA/cm² current density, with instantaneous output power density of 11.56 mW/m² when a load resistor of 0.5 MΩ was connected to the device. The rectified output charged commercially available capacitors and a light emitting diode. The architecture of the T-TENG included a rectangular glass substrate (due to its good transparency strength and machinability) and a PTFE and fluorine-doped tin oxide electrode. The fabrication process was simple and did not require any sophisticated equipment. PTFE has been selected due to its hydrophobic nature, resulting in self-cleaning, anti-sticking, and de-icing. It was also critical for TENG performance. Transmittance of the devices was tested using UV–VIS spectroscopy and the results were as follows: 83.41% for the FTO glass substrate, 85.24%, 87.177%, 86.98%, and 87.41%, respectively for the 3 μm, 2 μm, 1.5 μm, and 1 μm PTFE films. Additionally, PTFE film possesses antireflection properties as a coating. To power the T-TENG water from a faucet was used, with the flow rate being set at approximately 93 mL/s with 25 cm as the distance between T-TENG and the faucet. When resistors were connected to the T-TENG output, voltage increased from 0.1 to 2.75 V, for the load ranging from 10–5 MΩ, causing the output current to decrease from 10 μA to 0.07 μA. Since the devices usually required constant bias voltage or current, an integrated full-wave rectifying bridge (composed of a rectifying bridge, T-TENG, and a 22 μF capacitor) was used to transform the AC output to pulse output. The time needed to charge the capacitor to 0.7 V with T-TENG driven with the water stream was under 60 s. The high transparency made T-TENG suitable to be applied on buildings and vehicles to harvest rain energy beneficial for the construction of smart homes and car systems. For the tests, the authors [23] fabricated their own PTFE films, using diluted PTFE suspension, as the commercially available ones were not thick enough characterized by semi-transparency, when high transparency was needed for a highly transparent triboelectric nanogenerator.

Helseth et al. [24] described different dynamic modes in which it was supposed that water drops, when impacting a hydrophobic polymer, induce current in a metal electrode. It was found out that different modes followed different scaling laws, and that could be important in the specific harvesting systems analysed. Furthermore, water volume flow rates are the factors strongly determining the charging rate of a capacitor, the initial charging rate increases with volume flow rate, along with the individual droplets reach the metal electrode edge. The author [24] noted that the negative charge formed on neutral hydrophobic fluoropolymers upon water contact requires more comprehensive molecular modelling, systematic, and well-defined experimental studies of the contact electrification between water and a larger range of polymers. An electrode on the back surface of the polymer is used in the study conducted by the author [24] to measure charge separation occurring when water is removed from the polymer. This separation of charges depends on the type of the polymer and the conductivity of the liquid, among other factors. It was demonstrated that the charging rate of an external capacitor connected to the metal electrode through a rectifying bridge will depend strongly on the water volume rate over the polymer. In their experiments, the authors [24] used 75 mm-thick fluorinated ethylene propylene (FEP). They observed that a small amount of water does remain on the surface for some time and when removed the surface potential decreases as a result of ion removal. In the study, measurements were conducted on a freshly wetted surface to reduce the influence of the evaporation on the results obtained by the authors [24]. They studied four cases (current generating mechanisms) of drop movement as relates to the polymer surface and the electrode. When the drop was moving down the polymer surface (case 1), the electrical double layer generated charge equalled −50 mC/m², where the electrical charge input was only a fraction of the total charge density. Total electrical charge of double layer was significantly educin than the net electret charge. As the same droplet

moves over the edge of metal electrode at an angle of 44° with 180° (case 2), the total negative charge locates on the polymer surface, whereas the positive charge shifts to the metal electrode, indicating the current direction in the external circuit. As far as the charge is concerned, each segment is negatively charged (which means that the water drop is positively charged) and its magnitude varies from 0.5 nC to 1 nC. Most likely, it depends on the differences in droplet movement down the slope (rolling or sliding) and fluctuations or air-induced charging (i.e., gathering additional charge while displacing). The largest contact area occurred for the contact angle equal to 90°, while the smallest was for 180° [24] while the optimal results were recorded at θ = 109°. Cases 3 and 4 describe the droplet falling onto the edge of the electrode. Case No. 3 involves a drop bouncing on it, whereas, in case 4, the same drop was analysed in its most deflated state. When the total positive charge shifts to the metal electrode edge, a positive charge is allowed to flow into the metal electrode. To simplify, the current is mostly generated by flattening the droplet when it is spreading. The current was not observed in a case where the droplet made contact with the polymer far from the electrode edge. It was observed that the increment in the droplet velocity over the metal electrode increases the amount of current created. Detailed analysis provides evidence that the charge distribution in a spreading droplet is required while moving down the incline to change its charge distribution. Energy harvesting was done by connecting the metal electrode with the outer circuit composed of a bridge rectifier and a capacitor. The author [24] found that the voltage output appeared to be continuous since there were a lot of droplets randomly reaching the edge. It needs to be understood that the device built of the polymer coating and the metal electrode was only partially covered with the area of waterfall. Nonetheless, the gradual decline of the induced charge by passing droplets over the electrode was noted. It was due to the fact that when the polymer was already covered by a thin film of water, it disabled additional water flow which formed a new contact area, hence, additional charges did not form.

To summarise, Liang [23] presented a simple design for a triboelectric nanogenerator based on a PTFE polymer obtaining values comparable to piezoelectric devices. In Table 2, the electric parameters of the triboelectric devices are presented.

**Table 2.** Summary of selected electrical properties for triboelectric devices.

| Device Configuration | Voltage Output [V] | Current Density [μA cm$^{-2}$] | Power Density [μW cm$^{-2}$] | Ref. |
|---|---|---|---|---|
| PTFE/FTO/Glass | 10 | 2 | 1.156 | [23] |
| FEP/metal electrode | 3 | 0.026 | - | [24] |

## 3. Photoelectron Storage

In this section some application aspects of phosphorescent phosphors/long persistent phosphors (LPPs) are presented. Both materials exhibit a long persistent phosphorescence effect, also called afterglow, in which the material emits UV, VIS, or NIR after the excitation radiation has ceased. The LPPs applications ranged from decoration, displays, and dials to advanced application in biomedicine, life sciences, energy, clinical medicine, and environmental engineering. This is due to great contributions in the area of developing near-infrared (NIR) LPPs or nano-scale LPPs [25]. I concentrate on those aspects that allows for the use of those materials in energy harvesting.

The crucial aspects of afterglow mechanism LPPs can be divided into four parts:

(1)   excitation of the charged carriers;
(2)   storage of the charged carrier;
(3)   release of the charged carriers;
(4)   recombination of charged carriers.

Photon traps in the second stage are characterized by three factors: their concentration, depth, and type [25].

Currently, research describes myriad variety in the synthesis of new LPPs with different activators, hosts, and emission–excitation bands (UV–VIS–NIR) to meet the requirements of the market [26,27]; however, while designing the methodology of LPPs such aspects as electron band configuration, band theory, and redox are also crucial to understand the design–structure–properties relationships. The trapping of persistent phosphors involves the analysis of trap types, corresponding to lattice defects. Other factors to be considered are the concentration of effective traps and their depth, which is the time in which the captured carriers are released from the traps, as it impacts both the intensity and time of the afterglow. The capture and release process involves phenomena of formation of specific "tunnels" through a barrier that it normally could not overcome, or trapped by oxygen vacancies, allowing for the analysis of the retarded phosphorescence mechanism in oxide-based LPPs [28]. Figure 7 presents graphical representation of four different afterglow mechanisms.

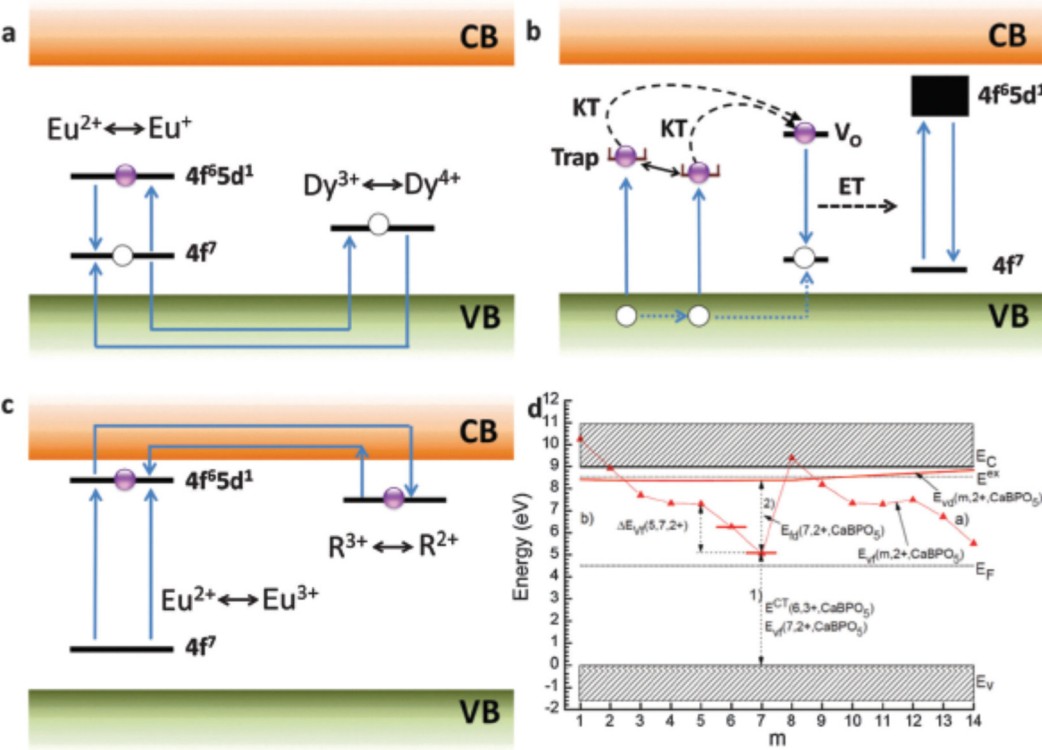

**Figure 7.** (**a**) Hole model: Afterglow mechanism proposed by Matsuzawa et al. for $SrAl_2O_4$:$^{Eu2+}$,$Dy^{3+}$. (**b**) Electron model: Afterglow mechanism proposed by Aitasalo et al. for $CaAl_2O_4$:$Eu^{2+}$,$Dy^{3+}$. (**c**) Persistent phosphorescence mechanism proposed by Dorenbos et al. for aluminate and silicate compounds. (**d**) Energy level scheme of the divalent lanthanides in $CaBPO_5$. Reproduced with permission from [25] with permission from The Royal Society of Chemistry.

The use of phosphorous materials implies an extended host area and activators. The involving hosts correspond to oxides (aluminates, silicates, stannates, phosphates, gallates, germinates, etc.), non-oxides (sulfides, nitrides, etc.), and organic carbon nanostructures, while the activators refer to rare earth ions and transition metal ions. LPPs classification is carried out two-fold, with regard to the materials and their operational wavebands. LPPs activators (activation ions) are luminescent centres. There are few of them and they include transition metal ions (such as $Cr^{3+}$, $Mn^{2+}$, $Mn^{2+}$, $Ti^{4+}$), rare earth ions ($Ce^{3+}$, $Pr^{3+}$, $Sm^{3+}$, $Eu^{2+}$, $Eu^{3+}$, $Tb^{3+}$, $Dy^{3+}$, $Tm^{3+}$, $Yb^{2+}$, $Yb^{3+}$), and $Bi^{3+}$. The electron store abilities of hosts can vary giving different trap contents. Two ways to form traps in the host are by the addition of intrinsic thermal or extrinsic defects (doping ions). Emission bands were studied, as far as visible and NIR range long-persistent phosphorescence were concerned. Bright green, blue, and white persistent phosphorescence was obtained over the years.

NIR long-persistent phosphorescence is induced by transition metals or rare earth ion doping [28–30].

If we consider, as a reference, absorption in conjugated polymers such as PFO-β, P3HT, PCDTBT, PTB7, Si-PCPDTBT, and PCPDTBT as active materials for organic solar cells. To increase the cell efficiency, we should look for materials that absorb different fractions of light other than the active material, but that also emit light at a specific wavelength. The chemical structures of the polymers are presented in Figure 8.

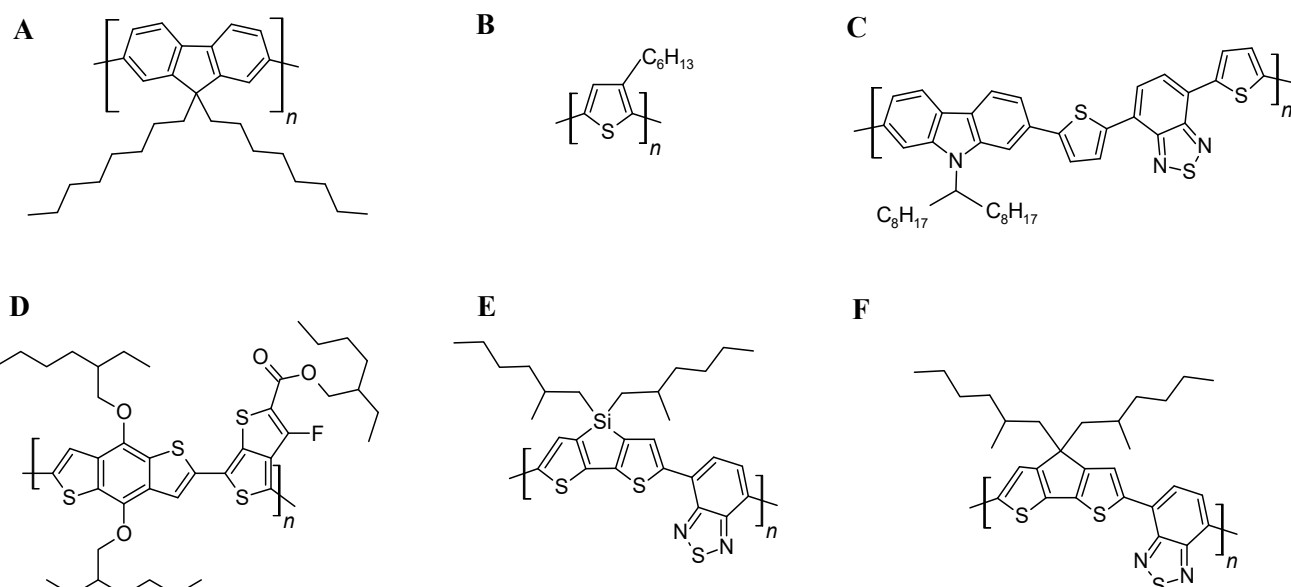

**Figure 8.** Chemical structure of polymers applied as active materials in organic solar cells: PFO-β (**A**), P3HT (**B**), PCDTBT (**C**), PTB7 (**D**), Si-PCPDTBT (**E**), and PCPDTBT (**F**).

The absorption maxima of the most commonly used organic materials are typically at a wavelength above 600 nm such as PTB7, Si-PCPDTBT, and PCPDTBT (see Figure 9) and the most suitable activators used in LPPs are rare earth ions. For example, binary oxide-based materials containing as host CaO, $SnO_2$, $Lu_2O_3$, $ZrO_2$ with activators such as $Eu^{3+}$, $Sm^{3+}$, $Tb^{3+}$ and sometimes co-dopant $Zr^{4+}$, $Sn^{4+}$, $Hf^{2+}$, $Ca^{2+}$ and $Sr^{2+}$. The most typical excitation band for this group of materials are bands between 254 and 300 nm or 900 nm, whereas the range of afterglow lies in the range between 550 and 720 nm [25,29–31].

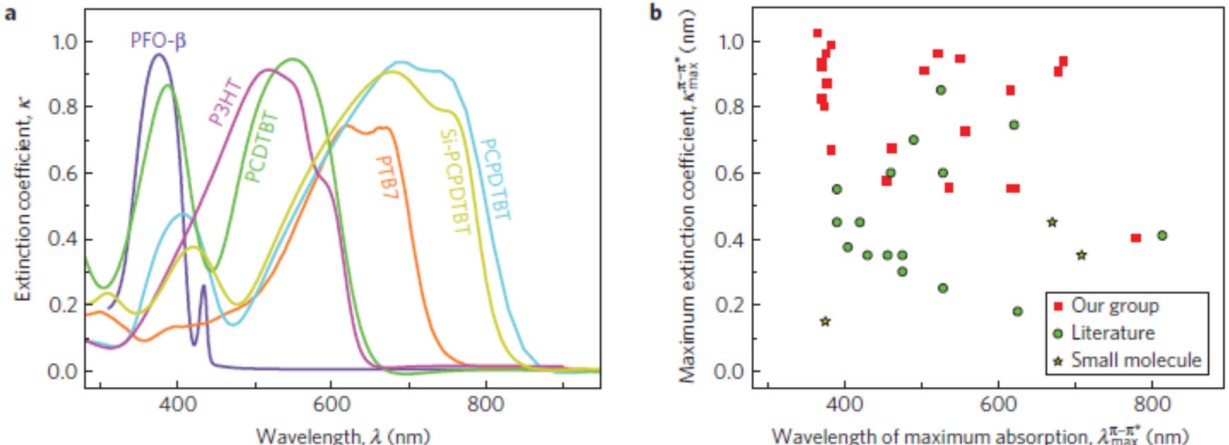

**Figure 9.** UV–Vis spectra (**a**) and wavelength of maximum absorption of conjugated polymers (**b**). Reproduced with permission from [32] Copyright © 2021, Springer Nature.

Similar relationships between absorption and emission bands as presented above, were observed for some sulphide- (i.e., CaS, SrS, $Y_2O_2S$), nitride- (i.e., $Ca_2Si_5N_8$, $BaAlSi_5N_7O_2$, AlN), and stannate- (i.e., $Ca_2SnO_4$, $Sr_3Sn_2O_7$, $CaSnO_3$) based phosphors with activators such as $Sm^{3+}$, $Eu^{2+}$, $Mn^{2+}$, and $Ti^{4+}$. The biggest advantage of those materials is longer afterglow time than in the case of binary oxide-based phosphors [25,33–35].

In the case of photovoltaic polymers such as PFO-β, P3HT, and PCDTBT the search for suitable supporting LPP material is more difficult since the absorption bands of those polymers are in the range of lower wavelength values below 600 nm and sometimes overlap with excitation bands of LPPs or the afterglow band does not correspond to absorption spectra of polymeric materials.

Alternatively, LPPs can enhance the capabilities of dye-sensitized solar cells, and for this purpose, phosphors should convert near UV from 250 to over 400 nm.

However, solar cells have low sensitivity to green light. The DSSCs structure was improved by forming a bottom layer composed of $SrAl_2O_4$:$Eu^{2+}$, $Dy^{3+}$ which was applied onto the $TiO_2$ nanoparticle layer. These were the working electrodes of DSSCs. The open-circuit voltage obtained was lower than that of the modified DSSCs. A working electrode with the phosphor layer improved the performance of the modified DSSC device. It resulted in a conversion efficiency 13% better than the device without this layer. The authors quote other work where, in their bifunctional structured layer, a 48% increase compared to the cell without $SrAl_2O_4$:$Eu^{2+}$, $Dy^{3+}$ was obtained. Using the same compound to manufacture composite films, conversion efficiency of a crystalline silicon photovoltaic increased as well, as it served as spectral sown-shifting. One of the vital conclusions of the paper which needs to be considered is the standardization of in the area of measurements and definitions of the properties of LPPs, as it would lead to performing reliable comparisons of the results obtained by different scientists on identical compositions [25].

LPP materials give a versatility to combinations of different host materials combined with dopant and co-dopant ions making it possible for adjustments in optical properties to almost whatever parameters are required. In Table 3, optical parameters of photoelectron storage are presented.

**Table 3.** Selected optical properties of phosphors materials.

| Host Material | Activator | Co-Dopant | Afterglow (nm) | Excitation Band | Afterglow Duration | Ref. |
|---|---|---|---|---|---|---|
| CaO | $Eu^{3+}$ | - | 592, 616 | 254 | 2 h | [25] |
| $ZrO_2$ | $Sm^{3+}$ | $Sn^{4+}$ | 550–700 | 254 | 900 s | [29] |
| $Lu_2O_3$ | $Tb^{3+}$ | $Ca^{2+}$, $Sr^{2+}$ | 543 | 270 | 20–30 h | [30] |
| $SnO_2$ | $Sm^{3+}$ | $Zr^{4+}$ | 550–700 | 254 | 900 s | [32] |
| CaS | $Eu^{2+}$ | $Tm^{3+}$ | 650 | Xe lamp | 1 h | [33] |
| SrS | $Eu^{2+}$ | $Pr^{3+}$ | 611 | 440 | 1000 min | [34] |
| $Y_2O_2S$ | $Eu^{3+}$ | $Mg^{2+}$, $Ti^{4+}$ | 590, 614, 627, 710 | 365 | 3 h | [25] |
| $Ca_2Si_5N_8$ | $Eu^{2+}$ | $Tm^{3+}$ | 500–750 | 420 | 200 min | [34] |
| $BaAlSi_5N_7O_2$ | $Eu^{2+}$ | - | 400–650 | 254 | 40 min | [34] |
| AlN | $Mn^{2+}$ | - | 570–700 | 254 | 1 h | [25] |
| $Ca_2SnO_4$ | $Sm^{3+}$ | - | 566, 609, 633 | 252 | 1 h | [25] |
| $Sr_3Sn_2O_7$ | $Sm^{3+}$ | - | 580, 621, 665, 735 | 267 | 1 h | [25] |
| $CaSnO_3$ | $Tb^{3+}$ | - | 491, 545, 588, 622 | 264 | 4 h | [25] |

## 4. "All-Weather" Solar Cells

### 4.1. Inorganic Solar Cells

To allow power generation power from both sunlight and rain precipitation energy, a harvesting structure combining a triboelectric nanogenerator (TENG) device and a solar cell was manufactured. An electrode made of a poly(3,4-ethylenedioxythiophene): poly(styrenesulfonate) (PEDOT:PSS) film was introduced to a heterojunction silicon (Si) solar cell integrated with a TENG. PEDOT:PSS was used to increase short current density.

Imprinted-polydimethylsiloxane (PDMS) was used as a triboelectric material, whereas a PEDOT:PSS layer served as an electrode. The output of the TENG was significantly improved due to a bigger area of contact between the raindrop and the imprinted PDMS. The values obtained were ca. 33.0 nA for short-circuit current and ca. 2.14 V for open-circuit voltage. To make use of renewable energy the TENG harvests solar energy during bright days and on cloudy days and when it rains raindrop energy is harvested. However, the structure of a hybrid energy harvesting system structure needs to be improved in order to reduce losses of the solar cell with reduced influence on performance of the TENG device. Thus, an electron hole-blocking layer, antireflection layer, structural modification of PEDOT:PSS, texturing the Si nanostructure, and surface passivation were applied and the Si/PEDOT:PSS solar cells obtained. In the paper, Liu et.al. [36] proposed a digital video disk pattern; a heterojunction Si solar cell was integrated with a single-electrode mode TENG. A PEDOT:PSS layer served as the common element of the two devices. In the case of the planar Si imprinted with PEDOT:PSS (called Si/imprint PEDOT:PSS) device the values obtained were the following (PCE, $V_{OC}$, $J_{SC}$, and FF, respectively): 13.6%, 0.628 V, 29.1 mA/cm$^2$, and 0.745. When textured Si/PEDOT:PSS was fabricated, the values differed and were as follows: PCE of over 12.6%, $V_{OC}$ of 0.612 V, $J_{SC}$ of 29.4 mA/cm$^2$, and FF of 0.702. The values for flat PEDOT:PSS on planar Si (Si/planar PEDOT:PSS) were as follows (PCE, $V_{OC}$, $J_{SC}$, and FF, respectively): 12.0%, 0.625 V, and $J_{SC}$ of 25.8 mA/cm$^2$, and FF of 0.746. $J_{SC}$ of 29.4 mA/cm$^2$ was due to light trapping. Textured Si's inferior diode property was due to a large Si surface/volume ratio. Furthermore the authors [36] studied the effective minority carrier lifetimes ($\tau_{ff}$) mapping measurement to evaluate the surface recombination velocity for different Si substrates, whose value for textured Si equals 10 µs, for textured Si/PEDOT:PSS equals 24 µs, for planar it was 37 µs, for planar Si/PEDOT:PSS the value was 62 µs, and that of the planar Si/imprinted PEDOT:PSS was 60 µs. To harvest raindrop energy, the TENG was built on the heterojunction Si solar cell. Vital information on imprinted PDMS is that film is hydrophobic, which is an advantage for a raindrop TENG device, as the rainwater will not wet the substrate. The vital characteristics of the TENG PDMS films were measured; the imprinted PDMS TENG device displayed superior electric performance as a consequence of a larger surface area. The influence of the angle (α) and the angle (β), angles between water dripping direction and device surface or imprinted structure, were studied for α and ranged from 15° to 75°. Average power first increased slightly and then decreased with the increase of α. As far as β is concerned, it was found that an increased angle leads to better output performance. The designed hybrid power system included TENG-generated current being transferred from AC via a bridge rectifier to DC. After rectification, current output of the TENG was ~24.2 nA. The high voltage TENG could compensate the shortcomings of the solar cell. The system underwent five recycling charging processes and indicated good stability and repeatability, with 1.74 mW m$^{-2}$ for the average value of power density. The system combines the possibility of high current output (solar cell) and high voltage output (TENG) (see Figure 10) [36].

The best performance was obtained for the cell with the imprinted layer of PEDOT:PSS without negative effect on the performance of the silicon solar cell during sunny days. The process of the texturized form of the polymeric layer had another beneficial input, meaning, its plaid role of antireflective coating. The best harvesting performance for the TENG module was observed for the incident angle of water set as 45°.

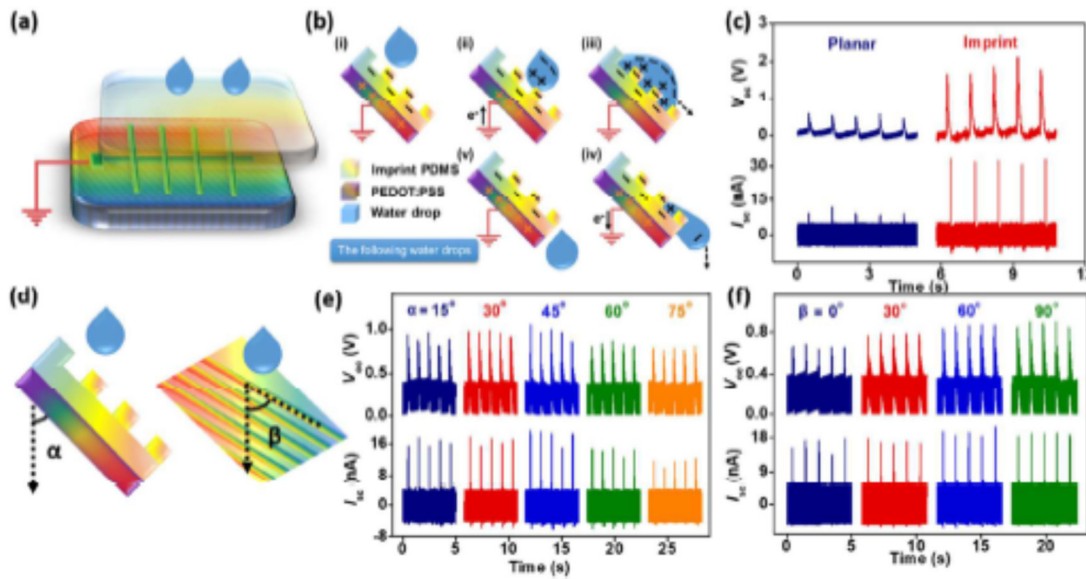

**Figure 10.** TENG device structure and general performance of TENG for harvesting mechanical energy from water drops. (**a**) Schematic illustration of a typical TENG structure; (**b**) Working mechanism of TENG; (**c**) $V_{oc}^{TENG}$ and $J_{sc}$ outputs of TENG with or without surface imprint; (**d**) Schematic illustration of angle $\alpha$ between water dripping direction and the device, and the angle $\beta$ between water dripping direction and imprinted structure; (**e**) $V_{oc}^{TENG}$ and $J_{sc}$ outputs of TENG under different $\alpha$; (**f**) $V_{oc}^{TENG}$ and $J_{sc}$ outputs of TENG under different $\beta$. Reproduced with permission from [36] Copyright © 2021, American Chemical Society.

### 4.2. Dye-Sensitized Solar Cells

Since electricity generation from the sun is zero at night and rainy weather, the authors decided to create an all-weather solar cell which would take advantage from both sun and rain. By hot-pressing graphene onto the rear side of the indium tin oxide/polyethylene terephthalate plastic substrate, a modified solar cell was built on an ITO layer. Due to the salt content in raindrops, when dropping onto a graphene surface, they reach the periphery forming an electric double-layer (EDL) pseudocapacitor at the interface of the raindrop and graphene. The shrinking drop releases electrons to the graphene, thus, charging the pseudocapacitor. It is not applicable for the fluorinated tin oxide (FTO) glass substrate to be used in solar cells due to the frangibility of glass, as well as its divergent nature; however, in all-weather solar cells it can be useful.

Zhang et al. [37] manufactured an all-weather solar cell on double-sided conductive glass. It was done via coating an ITO film on the rear surface (nonconducting side) of the commercially available FTO glass and then depositing a graphene film onto the solar cell. The deposition was carried out using the electrophoretic deposition method. In this type of solar cell, di-tetrabutylammonium cis-bis(isothiocyanato)bis(2,2′-bipyridyl-4,4′-dicarboxylato) ruthenium(II) (N719 dye), as photodye absorbs photons, release electrons to the conduction band of the $TiO_2$ nanocrystallite layer of the photocathode when the solar cell is irradiated from the anode by incident light. As far as electrodes are concerned, the authors [37] developed a CoNi alloy electrode in order to be more cost-effective in comparison to the Pt-based option. The efficiency (PCE) obtained equalled 8.16%. However, the increased triiodine monoanion to iodine redox process accelerates the recovery of N719 dye and, therefore, cell performance was improved to 9.14% when catalytic activity of ternary PtCoNi was used. To generate electricity on rainy days, the graphene monolayer needed to be placed on the top surface of the solar cell. In the visible-light region, the graphene monolayer was characterised by high optical transparency equalling ca. 92%. The efficiencies were 5.63% for solar cells with Pt, 6.96% in the case of the CoNi counter electrode, and 7.69% for the PtCoNi. Other characteristics for the CoNi were the following: PCE was 8.16%, $V_{oc}$ was 0.725, $J_{sc}$ was 17.68 m $Acm^{-2}$, and FF was 63.7%, whereas for Pt-CoNi the results were the following: PCE was 9.14%, $V_{oc}$ was 0.739, $J_{sc}$ was 18.48 mA $cm^{-2}$,

and FF was 66.9%. Good results were also obtained for the RuCoSe alloy electrode. PtCoNi and RuCoSe alloy electrodes exhibited maximal catalytic activities toward the redox electrolyte, hence, their maximal $V_{oc}$ and $J_{sc}$. Furthermore, an experiment has been conducted demonstrating that deionized water does not generate electrical signals when contacting the graphene surface. Since graphene is considered a Lewis base due to its cations, the $Na^+$ ion in raindrops behave as Lewis acids, absorbing the π-electrons from the graphene structure in terms of a Lewis acid–base interaction, thus, forming EDL pseudocapacitance at the interface of $Na^+$/graphene [38]. Other materials which are electron-enriched and can be applied to fabricate all-weather solar cells include polyaniline, polypyrrole, and alloys. The time interval between two droplets is called rain intensity (controlled by regulating injection velocity) and, apart from raindrop concentration, highly impact the signal output. Examples of electrical parameter dependence on simulated raindrop intensity is shown in Table 4.

When the case of acid rain was studied the authors hypothesised that a bi-triggering solar cell (see Figure 11) should yield higher current and voltage signals due to the greater ion concentration in acid rain. Despite promising results, the electricity generated from rain itself is much lower than required in practical applications, thus, a need for energy storage devices (i.e., all-weather solar cells) may be the optimal solution [37].

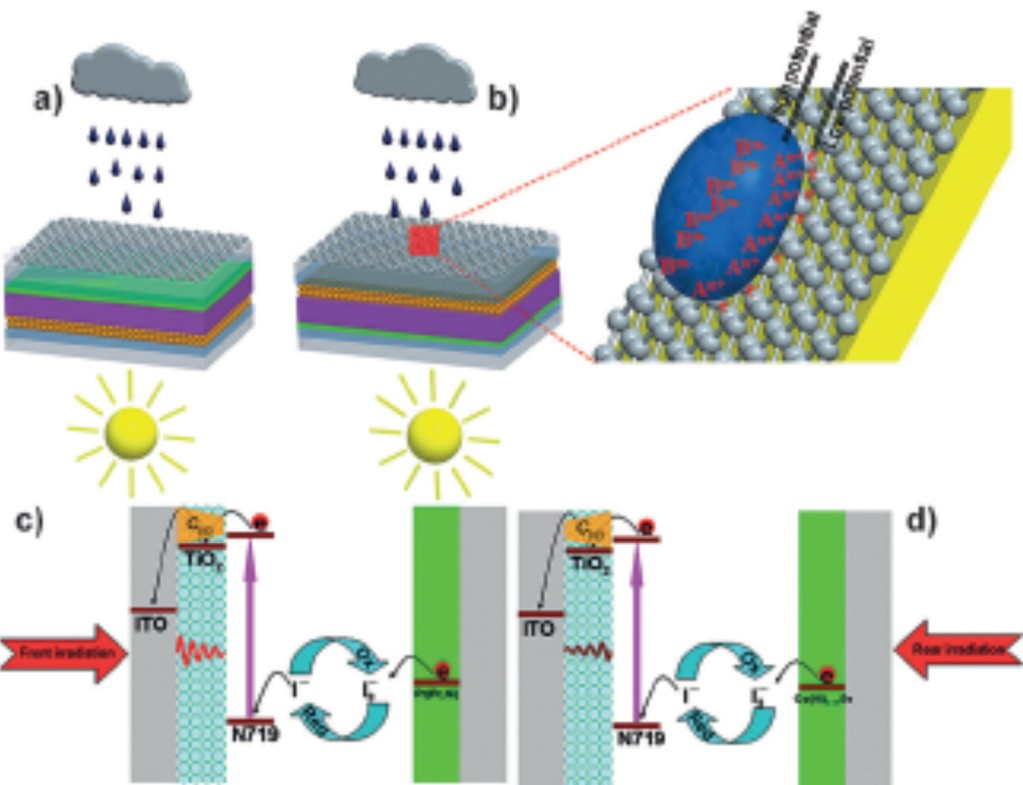

**Figure 11.** The bi-triggered flexible solar cell for recording photovoltaic performance under (**a**) front or (**b**) rear irradiation and measuring current and voltage signals by dropping raindrops (including $A^{n+}$ and $B^{m-}$ ions) onto rGO film. (**c**,**d**) The operational principle of the flexible solar cell under sunlight. The rGO film is deposited on the PET side and solar cell is assembled on the ITO side. The two electrodes for measuring current and voltage signals in the presence of raindrops are coated with silver paint and subsequently protected with ethylene vinyl acetate copolymer. The droplets are injected by a medical syringe by controlling injection velocity. Reproduced with permission from [5] © 2021 WILEY-VCH Verlag GmbH & Co. KGaA, Weinheim, Germany.

**Table 4.** The corresponding parameters for current, voltage, and power outputs produced by dropping simulated raindrops onto graphene electrode. Reproduced with permission from [37] Copyright © 2021, Royal Society of Chemistry.

| Injection Velocity [mL/h] Parameters | 40 [a] | 60 [a] | 80 [a] | 100 [a] | 200 [a] | 300 [a] | 100 [b] | 100 [c] | 100 [d] |
|---|---|---|---|---|---|---|---|---|---|
| Current [μA] | 4.9 | 2.1 | 1.6 | 1.4 | 1.2 | 0.9 | 1.9 | 1.4 | 1.2 |
| Voltage [μV] | 62.0 | 51.6 | 45.7 | 44.1 | 37.2 | 36.1 | 102.9 | 43.9 | 43.7 |
| Power [pW] | 303.8 | 94.8 | 73.1 | 53.7 | 44.1 | 33.0 | 183.1 | 45.2 | 31.9 |

[a] The concentration of NaCl aqueous solution is 0.6 M; [b] The concentration of NaCl aqueous solution is 1.0 M; [c] The concentration of NaCl aqueous solution is 0.5 M; [d] The concentration of NaCl aqueous solution is 1.0 M.

To achieve the objective of a combination of photovoltaics (conversion of sunlight energy) with other energy conversion working during night time or under dark conditions, Tang and his group [10] manufactured an all-weather solar cell by introducing long-persistence phosphors (LPPs) into m-TiO$_2$ photoanodes for the afterglow effect. LPPs are characterised by their ability to collect energy from ultraviolet and/or visible light and then produce afterglow light in a visible range at room temperature without the need for irradiation. A dye-sensitized solar cell (DSSC) is a third-generation solar cell consisting of dye-sensitized m-TiO$_2$ photoanode, a I$^-$/I$_3^-$ redox electrolyte, and a platinum counter electrode. It provided photovoltaic characteristics with PCE of 8.08%. The final *m*-TiO$_2$/LPP photoanodes were created by applying a coating with purple, blue, cyan, green, red or white-emitting LPP layer and these could be built into all-weather cells. Solar cells with LPPs and without LPPs were compared and the efficiencies were significantly greater and equalled 10.08% as a result of simulated light and afterglow effect. Moreover, it was found that the charge-transfer processes were improved because of the addition of an LPP layer. Furthermore, PCE was also increased and equalled 26.69%, 22.62%, 20.87%, 19.78%, 15.35%, and 3.02% for all-weather solar cells characterised by green, cyan, blue, purple, red, and white luminescence, respectively. Electricity could be generated in all of the above wavelengths. The incorporation of the above mentioned LPPs, the maximal incident photon to current efficiency (IPCE) values obtained equalled 84%, 78%, 71%, 56%, 47%, and 63%, respectively. However, $V_{oc}$, J$_{sc}$, and FF obtained under dark conditions were lower than those during daytime and the authors speculate that the reduced $V_{oc}$ and FF is due to the increased electron recombination reactions in the dark conditions, since the lower fluorescent light yields low electron density at TiO$_2$. It was found that the light ranging from 625 to 720 nm has low impact on excitation and, therefore, electricity output. However, the photoanode irradiated with white luminescence achieved broad emission in a range from ~415 to ~632 nm, with a maximum at 474 nm and other smaller maxima at 450, 468, 495, 511, 537, 586, 616, and 625 nm, in comparison with the double emission peaks for the red-luminescence anode (594 and 625 nm). This points to the possibility of emitting monochromatic light by the *m*-TiO$_2$/LPP photoanodes simply by absorbing sunlight. The authors [10] also studied the durability factor under dark and light conditions and found that the only darkening *m*-TiO$_2$/LPP photoanode is the one for purple and red luminescence, as a massive number of trapped electrons was released in the LPPs purchased by the authors. For the same TiO$_2$/LPP photoanodes with these above listed luminescence (in the order: green, cyan, blue, white, purple, and red), the following maximal emission intensities were the following: 171, 176, 160, 140, 136, and 134 mW cm$^{-2}$, respectively. As far as decay time (dark characteristics) is concerned, the authors [10] noted that all the relevant parameters ($V_{oc}$, J$_{sc}$, FF, and P$_{max}$) exhibited sudden reductions in the first 5 min and were almost constants following 55 min.

To summarize, the studied LPPs can capture and release incident light with wavelength >550 nm when illuminated by sunlight and subsequently afterglow at dark conditions. The six all-weather solar cells with different colours of luminescence and characterised by good long-term stability were manufactured and the maximized conversion efficiency of up to 26.69% was obtained in the total darkness (see Table 5). Figure 12 presents some experimental data of dark J–V characteristics [10].

**Table 5.** The photo (first part) and dark (second part) photovoltaic parameters for corresponding solar cells. η: power conversion efficiency; $V_{oc}$: open-circuit voltage; $J_{sc}$: short-circuit current density; FF: fill factor. Reproduced with permission from [10] © 2021 Elsevier B.V.

| Photoanodes | $V_{oc}$ (V) | $J_{sc}$ (mAcm$^{-2}$) | PCE (%) | FF (%) |
|---|---|---|---|---|
| $TiO_2$/LLP-green | 0.753 | 19.02 | 10.08 | 70.38 |
| $TiO_2$/LLP-cyan | 0.744 | 17.67 | 9.07 | 68.99 |
| $TiO_2$/LLP-blue | 0.732 | 17.29 | 8.62 | 68.11 |
| $TiO_2$/LLP-white | 0.725 | 17.25 | 8.39 | 67.09 |
| $TiO_2$/LLP-purple | 0.716 | 17.04 | 8.01 | 65.65 |
| $TiO_2$/LLP-red | 0.715 | 14.96 | 7.27 | 67.97 |
| $TiO_2$ | 0.703 | 16.36 | 8.08 | 70.25 |
| $TiO_2$/LLP-green | 0.343 | 0.247 | 26.69 | 67.66 |
| $TiO_2$/LLP-cyan | 0.326 | 0.226 | 22.62 | 67.11 |
| $TiO_2$/LLP-blue | 0.311 | 0.193 | 20.87 | 67.04 |
| $TiO_2$/LLP-white | 0.302 | 0.184 | 19.78 | 66.36 |
| $TiO_2$/LLP-purple | 0.298 | 0.138 | 15.35 | 65.77 |
| $TiO_2$/LLP-red | 0.157 | 0.065 | 3.02 | 63.05 |
| $TiO_2$ | 0 | 0 | 0 | - |

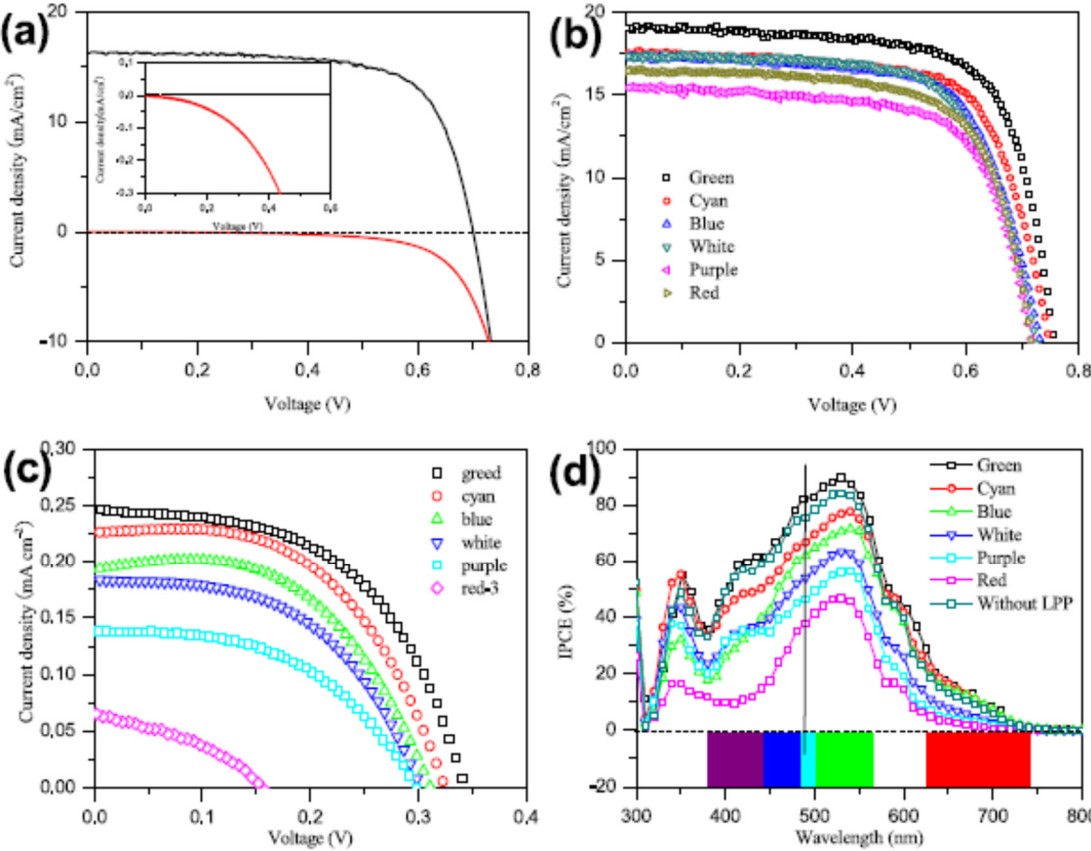

**Figure 12.** (**a**) Representative photo and dark J–V characteristics of the DSSC including a dye-sensitized $TiO_2$ anode, a $I^-/I_3^-$ redox electrolyte, and a Pt CE. The inset means the magnified dark J–V curve. (**b**) Characteristic photo J–V curves for all-weather DSSCs. An all-weather DSSC device comprises a dye sensitized $TiO_2$/LPP photoanode, a $I^-/I_3^-$ redox electrolyte, and a Pt CE. (**c**) The J–V curves for all-weather DSSCs recorded under completely dark conditions. (**d**) IPCE plots of the DSSCs with and without LPP phosphors. All photo J–V curves are measured under simulated sunlight (AM1.5, 100 mW cm$^{-2}$). The dark J–V curves are recorded in atmosphere with light intensity of 0 mW cm$^{-2}$. Reproduced with permission from [10] © 2021 Elsevier B.V.

Tang et al. [6,39] presented a preliminary study on all-weather solar cells capable of harvesting both the solar and raindrop energy via the incorporation of a graphene-based layer with a solar cell. The mechanism behind graphene adsorption from liquids to form an electrical double layer (EDL) (separation of $\pi$-electron and cation) at the interface of the graphene and ionic liquid was as follows:

(i) When they spread to the periphery and form this EDL pseudocapacitance they drag the electron transfer, and the front of the raindrops are charged;

(ii) Afterward, the raindrops shrink and release electrons to the graphene and discharge the pseudocapacitance, and the repeated charging and discharging processes yield current and voltage.

This capacitive response during the shrinking and spreading was investigated using cyclic voltammogram characterisation. Graphene (conducting electrons) and carbon black with a polytetrafluorethylene (PTFE) insulator to construct a graphene–carbon black conducting composite ((G-CB)/PTFE) was fabricated in order to modify graphene dosage, improving film-forming ability. Carbon black serves as the compatibility improver for the graphene/ PTFE mixture. In this construction, graphene is responsible for electron migration by the $\pi$-electron system. By combining a solar cell with a G-CB/PTFE film (characterised using Raman, XRD, FTIR, TGA, SEM, and TEM), an all-weather solar cell was created to harvest energy from sunlight and rain. Photosensitive N719 dye, which absorbs photons, releasing electrons to the FTO layer along porous $TiO_2$ pathways, was used. Although the solar cell with a Pt electrode can yield a PCE of 7.23%, it is a costly option for mass production and that is why a PtNi alloy counter electrode was fabricated, and an efficiency of 9.80% was obtained (see Figure 13). The voltage and current response to simulated rain are presented in Figure 14.

The J–V curves were recorded when simulated raindrops (NaCl aqueous solution) were being dropped onto G-CB/PTFE conducting films, whereas pure, deionized water produced no current. To describe the adsorption and transportation of the charges (see Figure 15), electrochemical impedance spectroscopy (EIS) was performed.

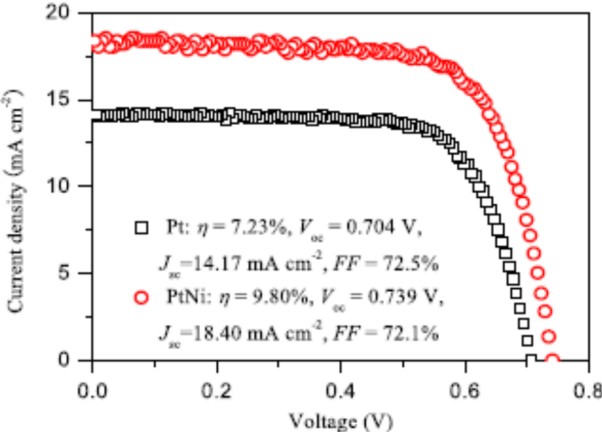

**Figure 13.** J–V curves of the all-weather solar cells under sunlight irradiation (AM1.5). $\eta$: photoelectric conversion efficiency; $V_{oc}$: open-circuit voltage; $J_{sc}$: short-circuit current density; FF: fill factor. Reproduced with permission from [39] © 2021 Elsevier Ltd.

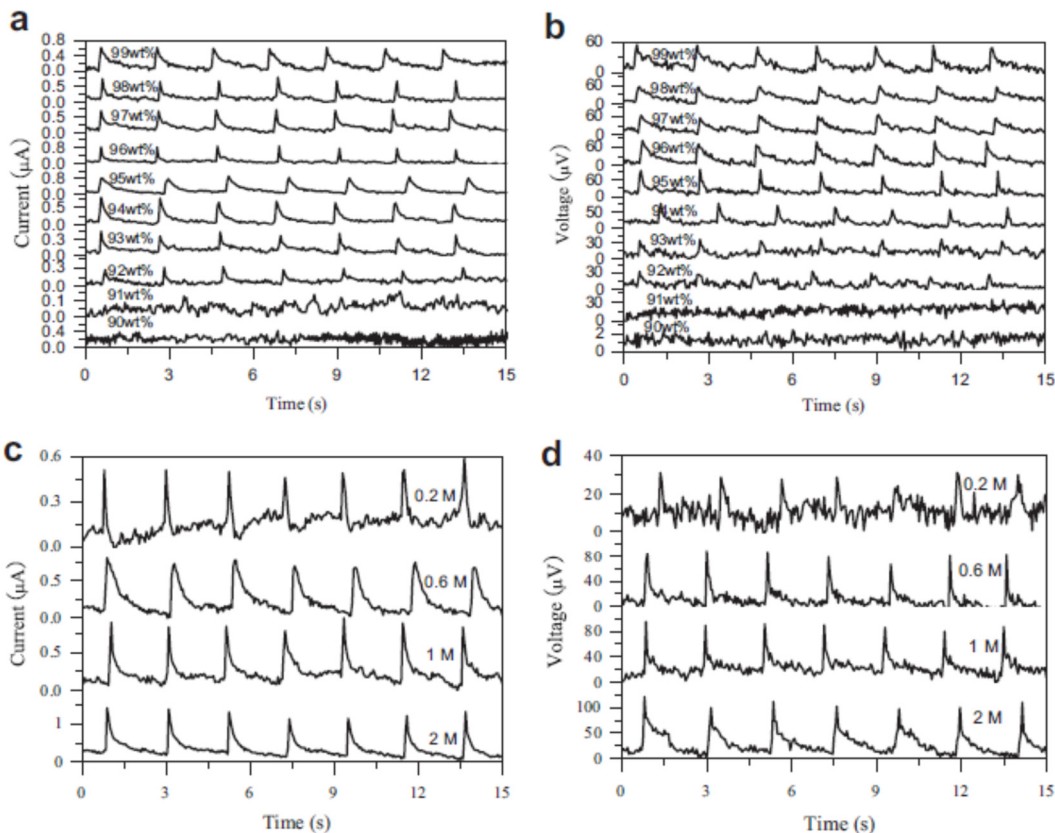

**Figure 14.** (**a**) Current and (**b**) voltage signals produced by dropping simulated raindrops (0.6 M NaCl aqueous solution) on G-CB/PTFE electrodes with various G-CB dosages of all-weather solar cells. (**c**) Current and (**d**) voltage signals produced by dropping simulated rain droplets (NaCl aqueous solutions at different concentrations) on 95 wt% G-CB/PTFE conducting electrode of the all-weather solar cell. Reproduced with permission from [39] © 2021 Elsevier Ltd.

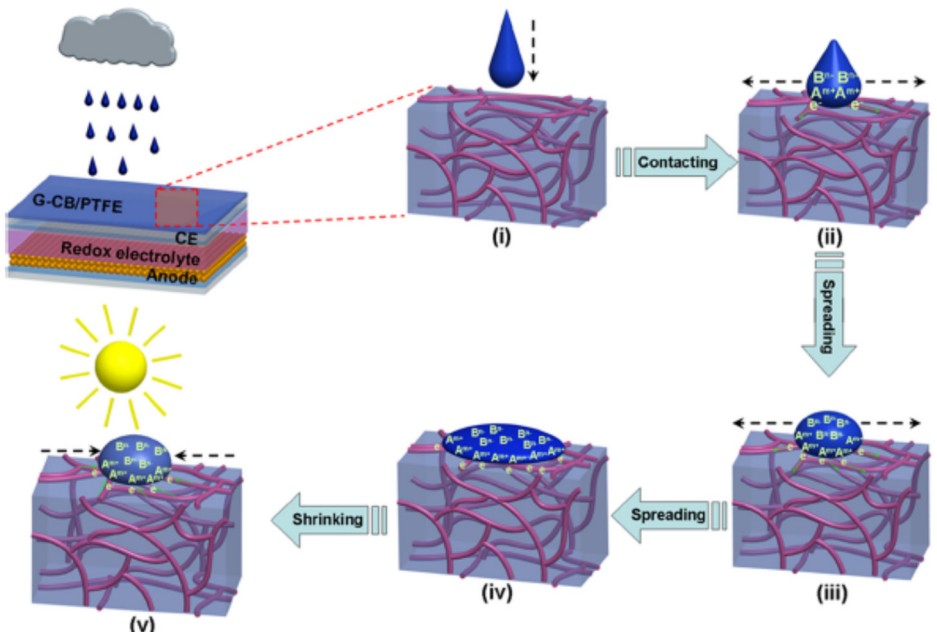

**Figure 15.** The charge adsorption and transportation pathways during the spreading/shrinking processes for a raindrop on G-CB/PTFE conducting composite of all-weather solar cell. Reproduced with permission from [39] © 2021 Elsevier Ltd.

Furthermore, the $Na^+$ concentration influence on electric signals was studied: NaCl aqueous solution concentration increased from 0.2 to 2 mol $L^{-1}$. The current and voltage values for 2 M NaCl solution were 1.14 µA and 100.10 µV, which is a significant increase from the values for the 0.2 mol $L^{-1}$ solution (0.35 µA and 25.65 µV, respectively, see Figure 14). The greater the cation concentrations (such as in the case of acid rain) involved more π-electrons involved in improving pseudocapacitance and, thus, electrical response. The authors [39] also wanted to cross-check the G-CB conducting pathway formation, so they used a polyacrylate/sericite composite because of the sericite distinctive polarized light behaviour and the formation of interconnected networks within the composite film. By studying the electrical percolation, the authors [39] found that the beneficial wt% of G-CB is 95, as the 95 wt% G-CB/PTFE composite electrode showed a lot of promise, as far as rain energy generation was concerned. Rainfall intensity (injection intensity) is another factor impacting the current and voltage signals, as they decrease with increased rain intensity. This is because, low injection velocity causes an electron recombination with the $Na^+$ in the previous droplet when the next raindrop falls at high velocity. This reduces charging pseudocapacitance. Furthermore, since positively charged species and charge number exert vital impact on the current and voltage signals, other solutions have been studied. The cell characteristics increased to 1.14 µA and 89.31 µV for 0.6 M LiCl solution; however, changing a lithium cation for potassium causes a reduction to 0.61 µA and 33.89 µV. The results obtained for magnesium chloride and calcium chloride aqueous solutions gave better results than that for sodium and potassium chloride, respectively. Lastly, pH and temperature impacting current and voltage were studied, and the findings were that current and voltage outputs increase with increased pH value and temperature. That conclusion is key for manufacturing all-weather solar cells for those regions of the globe with high temperatures and acid rain.

It is well known that solar cells can only be excited by sunlight on sunny days and do not function (or function very weakly) on rainy days. To address this issue, professor Tang's group [5] developed a new solar cell composed of an electron-enriched electrode (rain energy harvesting) with a DSSC for photoelectric conversion. Electricity generation was conducted by raindrops falling on graphene film, as rain contains both the positively ($Na^+$, $Ca^{2+}$, and $NH_4^+$) and negatively charged ions. When raindrops drop onto the graphene surface, the positively charged ions are adsorbed onto this surface by Lewis acid–base interactions to drive electron migration. Thus, electrical π-electron/cation double-layer pseudocapacitors are created (see Figure 16).

The authors [5] fabricated a bi-triggered DSSC with $PtNi_3$ alloy, which provided photoelectric conversion efficiency equalling 6.53% and noted that the use of Pt-Ni alloy can remarkably increase electrolyte contact and enhance the catalytic activity. This cell can also generate energy from the rear, counter electrode, and side by the application of $Co_{0.85}$, Se, and $Ni_{0.85}$ Se with high optical transparency [40,41]. Efficiencies of 4.26% and 4.09%, respectively, were obtained. During rain periods, proposed new solar cells could be covered with a reduced graphene oxide (rGO) film, allowing for current and voltage production under simulated rain (0.6, 1 and 2 M NaCl aqueous solution). The reduction of the lateral distance between the falling point and electrode to 7.58 and 4.52 mm increases the current intensity to 0.33 and 0.54 mA, respectively. Another factor influencing the electrical signals is the time interval between two droplets (controlled by injection velocity). The results obtained by the authors were the following: 20 $mLh^{-1}$ (ca. 0.49 mA, ca. 109.26 mV, ca. 54.19 pW), >50 $mLh^{-1}$ (ca. 0.37 mA, ca. 44.48 mV, ca. 20.63 pW), >80 $mLh^{-1}$ (ca. 0.17 mA, ca. 31.89 mV, ca. 5.12 pW). The third factor is the high dependence of current and voltage on the ion concentration, the higher the $Na^+$ concentration (2 M vs. 1 M), the more enhanced the current and voltage. The authors [5] demonstrated that those electrons which had not formed the π-electron/cation pseudocapacitance could further interact with $Na^+$ ions when the concentration is high. The authors [5] also focused on the long-term stability and they highlighted their supremacy over the actual value of the signal. Following the repeated operation of this new type of solar cell, 88.8% of initial current

and 52.3% of voltage remained after the repeated dropping of 0.6 M NaCl solution on the rGO during the 1000 s period with 4 s intervals. The deterioration is most likely due to rGO susceptibility to NaCl solution, namely the increased surface wettability or increasing the interfacial resistance. Focusing on the future design improvement and stability, the authors recommend discarding rGO in favour of a highly hydrophobic graphene film with compact stacking in order to improve this particular characteristic.

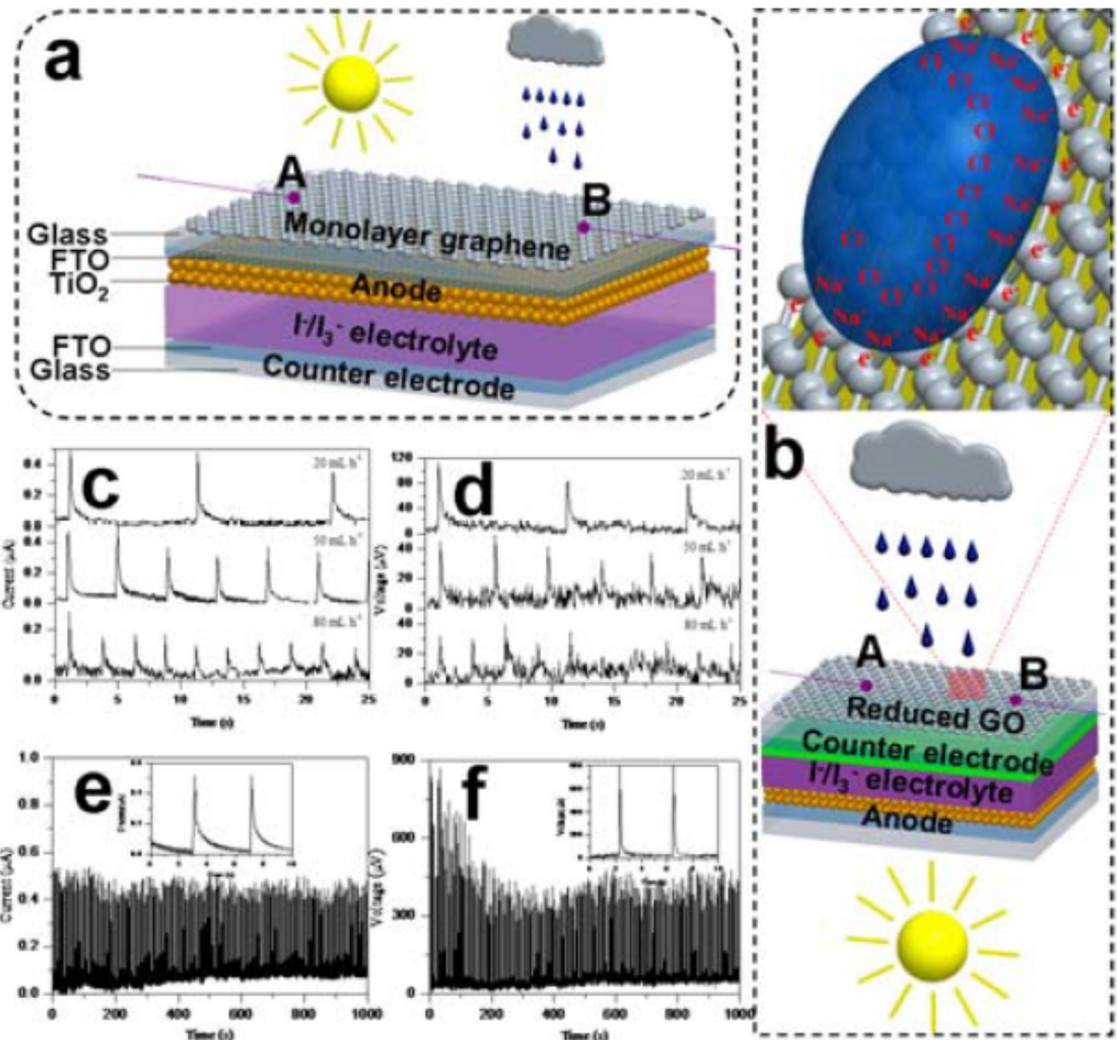

**Figure 16.** Schematic of the sun- and rain-enabled all-weather solar cell devices by covering (**a**) a monolayer graphene film on the top surface of photoanode as well as (**b**) reduced GO on back surface of a CE. The (**c**) current and (**d**) voltage outputs as well as long-term (**e**) current and (**f**) voltage stability by dropping simulated rain on b-type architecture. Reproduced with permission from [6] © 2021 WILEY-VCH Verlag Gmbh & Co. KGaA, Weinheim, Germany.

Tang et al. [5] explained the process of generating electricity from graphene when ions (from raindrops) interact to form electron/cation EDL pseudocapacitors based on Yin et al. [38]. During rain, current and voltage signals persistently yield charging/discharging cycles, and when sunlight illuminates the device, the photoelectric conversion processes follow the DSSCs principles. When capacitance was studied (using cyclic voltammetry) as a relationship between the rainwater contact area and graphene surface, it was found that during the spreading process it increased from zero to a maximum, returning to zero at the shrinking stage. Different methods of all-weather solar cells were presented, including inkjet printing. This method is characterized by simplicity, low cost, and capability of being produced on a mass scale, and it consists of a solute dissolved or otherwise dispersed in a solvent to form inks. The example provided is a conducting coat-

ing, a graphene-carbon black/polytetrafluorethylene (G-CB/PTFE), where the graphene allows for electron migration and the carbon black compatibility improves the graphene and PTFE matrix. Apart from graphene, some alloys (such as platinum with transition metals) can be used in all-weather solar cells, and the results obtained in the form of electric signals are at the same level as for graphene. Another approach includes long-persistence phosphors (LPPs) being coated onto a *m*-$TiO_2$ layer to produce electricity in dark conditions without reducing photo efficiency. LPP (either purple, blue, cyan, green, red, or white) is a phenomenon in which ultraviolet, visible, NIR, or infrared spectral regions are emitted for a specific period of time after irradiation has ceased. In the absence of light, the photoactive dyes are irradiated by the afterglow from LPP phosphors, yielding as 26.69% efficiency for the green-emitting LPPs devices. The N719 dye can be substituted with converted carbon quantum dots (CQDs) [42]; however, the efficiencies yield much lower values than those of the high-efficiency solar cells due to weak affinity of the CQDs and $TiO_2$ surface, hence, the need for modifications of the band energy structure.

Basic DSSC modules obtain PCE up to 11% and is similar for all bi-triggered setups. The biggest difference lies in the rain harvesting system and system with LPP. The biggest disadvantage of the rain harvesting system are rain-dependent factors such as rain direction and force of the impact, hence, the obtained energy is rather punctual and gives small values of current and voltage under a load. Furthermore, the systems with a photon storage layer gives a continuous response in dark conditions with values comparable to those obtained under illumination; however, the storage capacity is very limited and the produced current drops with time. For comparison reasons, in Table 6 are presented photovoltaic parameters of all-weather solar cells.

**Table 6.** The performance summary of rain-powdered all-weather solar cells.

| Electrodes | PCE (%) | Current (µA) | Voltage (µV) | Ref. |
|---|---|---|---|---|
| rGO | 6.53 | 0.49 | 109.26 | [5] |
| Electrophoretic graphene | 9.14 | 4.9 | 62.0 | [37] |
| Monolayer graphene | 7.69 | 0.66 | 61.8 | [37] |
| G-CB/PTFE | 9.8 | 0.78 | 77.52 | [39] |
| Pt | 6.75 | 0.11 | 0 | [6] |
| $PtNi_5$ | 10.38 | 1.13 | 27.76 | [6] |
| $PtFe_5$ | 8.83 | 0.67 | 22.55 | [6] |
| $PtCo_5$ | 8.47 | 3.90 | 115.52 | [6] |
| $PtCu_5$ | 8.22 | 3.18 | 76.39 | [6] |
| $PtMo_5$ | 6.69 | 2.18 | 57.60 | [6] |
| Ni | 0.93 | 0 | 0 | [6] |
| $TiO_2$/LPP-green (dark) | 26.69 | 0.247 | 0.343 V | [10] |

## 5. Environmental Aspects in All-Weather Solar Cells

The environmental aspect, in terms of everything that is produced by humans, is an important matter that has been highlighted in the last decade [43,44]. Current interest in renewable energy, especially organic solar panels, and perhaps in the near future, all-weather solar cells, brings a question: what should we do with the energy harvesting devices when they are no longer needed for use? There are several aspects of eco-friendliness to be considered:

(i)   harmless, safe, and cheap components of designed materials [45];
(ii)  simplicity and "greenness" of synthetic pathways [44];
(iii) long durability of the product;
(iv)  material/device "afterlife" recyclability and disposal [46].

Current research on all-weather solar cells, use materials and technology already available. Apart from the typical materials used in the construction of solar cells, the additional components mainly include materials such as: perfluorinated polymers (i.e., PVDF, PVDF-TrFE, PTFE, FEP), conducting polymers (PEDOT:PSS), carbon materials (i.e.,

graphene, graphene oxide, reduced graphene oxide, carbon black), and inorganic materials (i.e., $TiO_2$, ITO, FTO).

The most harmful product from the examples mentioned above are fluorinated polymers because of the products of their decomposition, such as hydrogen fluoride, carbonyl, tetrafluoroethylene, and perfluoroisobutylene, that are very toxic, and even short exposure can cause neurological or respiratory problems for living beings. The suggested method of waste management of fluorinated polymers is their recycling by re-extrusion, filtration, and purification [47].

"Cradle-to-cradle", in the context of photovoltaic devices, is the new policy established within the 7 Framework Program that encourages change in the current policy and makes recommendations:

(i)    support for (current) reducing the administration regarding the waste legislation for PV modules;

(ii)   research support, as the rules in the near future will include all-weather solar cells [48].

## 6. Conclusions and Future

The concept of all-weather solar cells, where power is obtained both from the impact of raindrops and solar energy or in the dark after the emission of absorbed energy, is an interesting idea that could increase the efficiency of the currently used solar cell. Systems with rain harvesting modules would be an especially beneficial option in areas with heavy rain fall all over the globe. Moreover, the inexhaustible water-related energy sources from waterfalls, rainwater, and ocean waves are not limited by the time of day (as opposed to solar energy), weather, or climate. It is difficult to accurately compare the various harvesting methods since the magnitude of generated energy differs significantly. The best results for specific harvesting methods were 213 $\mu$W cm$^{-2}$ (2076 $\mu$J) for 7.6 V generated energy from the piezoelectric device, and 1.156 $\mu$W cm$^{-2}$ (for current 2 $\mu$A cm$^{-2}$) for 10 V generated energy in the case of the triboelectric device.

Regarding rain energy, there are two types of energy carried by flowing water: mechanical energy (piezoelectric) and electrostatic energy (resulting from the tribo-charges formed during the changes in contact upon processing the force of air or other materials). As it was presented in the section of mechanical energy conversion, the architecture of a device matters significantly. Whereas the simples PVDF membrane can produce a theoretically very small amount of energy in the order of $10^{-19}$ $\mu$W cm$^{-2}$., however, it can be used as a component together with other materials. The more engineered devices gave reasonable parameters either in terms of obtained potential around 7 V or acceptable power density of 213 $\mu$W cm$^{-2}$. It could be interesting to use a wider approach to harvest mechanical energy, especially for a building facade, not only from falling rain drops but also vibration coming from wind movements. The triboelectric energy comes with a value of a single $\mu$W cm$^{-2}$, depending on the device architecture, tilde angle, and the composition of the rain drop. When we compare the energy harvesting from the rain in a mechanical or electrostatic manner, it can be noticed that the mechanical force gave higher input than the electrostatic forces. It demonstrates that in terms of energy from natural events like precipitation, the level of energy conversion is evidence that these technologies for now cannot serve as a main energy source but can be used to support other devices like solar panels. For the moment, this can only offer limited aid in increasing the harvested energy further.

Moreover, PTFE, the most common material, characterised by a combination of a low dielectric constant, low friction coefficient, high mechanical strength, high stability, and superb plasticity, can be easily used to construct all-weather solar cells.

Another potential technology is photoelectron storage, which assumes that a portion of light of a specific wavelength could be absorbed by LLP material to be later irradiated back with a wavelength adequate for a fotoactive material to generate current. If we consider LLPs materials as an additional component to conventional solar cells, it gives enormous possibilities for design and preparing a customized material suitable for the specific combination of photoactive materials used in a photovoltaic device. Moreover,

the variety of LPPs synthesis methods in a solid state allows for the use of known and cheap technology to produce those materials. The biggest challenge in the use of LLPs is their synchronization with the absorption range of photoactive dye: it needs to collect only those sections of light spectra "wasted" and to transform them into a specific light band (afterglow) that can be used during the dark hours by a PV device. In the case of the most commonly used photoactive materials, the absorption band of active layer ranges from 350–820 nm, so the best LLP material would be $Lu_2O_3$ activated with $Tb^{3+}$ and co doped $Ca^{2+}$, $Sr^{2+}$ excited at 254 nm providing 20–30 h of afterglow at 543 nm [30]. These parameters perfectly fit materials such as P3HT (absorption maximum at approx. 520 nm) or PCDTBT (absorption maximum at approx. 560 nm) used for organic PV construction.

Another material that can be used in all-weather solar cells is graphene with all its modification variations. The introduction of graphene electrodes, where power is obtained both from the impact of raindrops and solar energy, increase the solar to electric conversion rate to 22%.

Summarizing, solar cells for all weather is the future trend that will be explored further, since at some point researchers will reach the top boundary of light-to-electricity conversion, and to satisfy increasing needs for more electrical power, will need to combine different energy harvesting technologies to build hybrid systems.

**Funding:** Polish National Centre of Research and Development (TECHMATSTRATEG1/347431/14/NCBR/2018).

**Acknowledgments:** The author is grateful to Agnieszka Iwan for the inspiration and scientific discussion and to Patrycja Wojcieszyńska for help with language translation and correction.

**Conflicts of Interest:** The author declares no conflict of interest.

## Abbreviations

The following abbreviations are used in this manuscript:

| | |
|---|---|
| LLP | long persistent phosphors |
| PVDF | polyviniliden fluoride |
| PVDF-TrFE | poly[(vinylidenefluoride-co-trifluoroethylene] |
| T-TENG | transparent triboelectric nanogenerator |
| PTFE | polytetrafluoroethylene |
| GO | graphene oxide |
| rGO | rGO, reduced graphene oxide |
| FEP | fluorinated ethylene propylene |
| PZT | lead zirconatetitanate |
| LPPs | phosphorescent phosphors/long persistent phosphors |
| PFO-β | poly(9,9-dioctylfluorene) β-phase |
| P3HT | Poly(3-hexylthiophene-2,5-diyl) |
| PCDTBT | Poly[N-9′-heptadecanyl-2,7-carbazole-alt-5,5-(4′,7′-di-2-thienyl-2′,1′,3′-benzothiadiazole)] |
| PTB7 | Poly [[4,8-bis[(2-ethylhexyl)oxy]benzo[1,2-b:4,5-b′]dithiophene-2,6-diyl][3-fluoro-2-[(2-ethylhexyl)carbonyl]thieno[3,4-b]thiophenediyl ]] |
| Si-PCPDTBT | Poly[(4,4-bis-2-ethylhexyl)-dithieno[3,2-b:2′,3′-d]silole)-2,6-diyl-alt-(2,1,3-benzothiadiazole)-4,7-diyl] |
| PCPDTBT | Poly[2,6-(4,4-bis-(2-ethylhexyl)-4H-cyclopenta [2,1-b;3,4-b′]dithiophene)-alt-4,7(2,1,3-benzothiadiazole)] |
| TENG | triboelectric nanogenerator |
| PDMS | polydimethylsiloxane |
| DSSC | dye-sensitized solar cell |
| DSD | drop site distribution |
| EDL | electric double-layer |

| ITO | indiumtin oxide |
|---|---|
| FTO | fluorinated tin oxide |
| PEDOT:PSS | poly(3,4-ethylenedioxythiophene): poly(styrenesulfonate) |
| G-CB | graphene-carbon black |
| CQD | carbon quantum dot |
| $V_{OC}$ | open circuit voltage |
| $J_{OC}$ | open circuit current |
| PCE | power conversion efficiency |
| FF | fill factor |

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
