# Peer review of "Bi-Triggering Energy Harvesters: Is It Possible to Generate Energy in a Solar Panel under Any Conditions?"

_energies, doi:10.3390/en14185796_

Round 1

Reviewer 1 Report

The aper provides an extensive literature review on energy harvester experiments and technologies, with sufficient depth of analysis and references. Some comments are provided below:

Comments:

  • Could authors clarify the sentence “A maximum power of 2.231×10-29 W was obtained from a 1 m2 area was of a polyvinylidene fluoride (PVDF) membrane during a heavy thundershower”, considering the Earth surface not being covered with water is around 149 x 10^12 m2; that system will be able to produce less than 10^-17 W, how can that figure even exist? Or being measured? Even considering all the Solar System surface (no only the planets surface but the space voids) the system will produce around 240 W. From engineering point of view there is no sense in those studies; please justify their validity and real applications
  • It would be highly recommended adding a  table at the end of the paper summarising main characteristics; pros & cons; applications of the proposed systems 

Format Comments

  • Use Energies Template and publishing style recommendations
  • Check carefully manuscript writing. Professional English proofreading is highly recommended.

Author Response

# reviewer 1

  1. Could authors clarify the sentence “A maximum power of 2.231×10-29 W was obtained from a 1 m2 area was of a polyvinylidene fluoride (PVDF) membrane during a heavy thundershower”, considering the Earth surface not being covered with water is around 149 x 10^12 m2; that system will be able to produce less than 10^-17 W, how can that figure even exist? Or being measured? Even considering all the Solar System surface (no only the planets surface but the space voids) the system will produce around 240 W. From engineering point of view there is no sense in those studies; please justify their validity and real applications.

Ad.1. I am grateful for the comment. The reason for putting this citation in this review was to show that PVDF membrane - a commonly used element to construct any devices, can be used to generate energy. It was meant to interest reader in the phenomenon and present the very beginning of form where the idea derived.

  1. It would be highly recommended adding a  table at the end of the paper summarising main characteristics; pros & cons; applications of the proposed systems 

Ad.2. I am thankful for the comment. I extended the conclusion of the article adding a summary of all sections:

  1. Conclusions and future

The concept of all-weather solar cells, where power is obtained both from the impact of raindrops and solar energy, or in dark after emission of absorbed energy in and interesting idea that could increase the efficiency currently use solar cell. Systems with rain harvesting modules would be especially beneficial option in arears with heavy rain fall all over the globe. Moreover, the inexhaustible water-related energy sources from waterfall, rainwater and ocean waves are not limited by the time of day (as opposed to the solar energy), weather or climate. It is difficult to accurately compare the various harvesting methods since the magnitude of generated energy differs significantly. The best results for specific harvesting methods are: for 7.6 V generated energy was 213 µW cm-2 (2076 µJ) coming from piezoelectric device and for 10 V generated energy was 1.156 µW cm-2 (for current 2 µA cm-2) in the case of triboelectric device.

Regarding rain energy, there are two types of energy carried by flowing water: mechanical energy (piezoelectric) and electrostatic energy (resulting from the tribo-charges during the contact electrification process with air and other materials). As it was presented in the section of mechanical energy conversion, the architecture of an device does matter significantly. Whereas the simples PVDF membrane can produce theoretically very small amount of energy in order of 10-19 µW cm-2., however it can be used as a component together with other materials. The more engineered devices gave reasonable parameters either in terms of obtained potential around 7 V or acceptable power density 213 µW cm-2. It could be interesting to use wider approach to harvest mechanical energy, especially for building facade, not only from falling rain drops but also vibration coming from wind movements. The triboelectric energy comes with a values of single µW cm-2, depending on the device architecture, tilde angle and the composition of the rain drop. When we compare the energy harvesting from rain in mechanical or electrostatic manner, it can be noticed that mechanical force gave higher input than the electrostatic forces. It only demonstrates that in terms of energy form natural events like precipitation, the level of energy conversion put into evidence that these technologies for now cannot serve as a main energy source but can be used to support other devices like solar panel. For the moment, this kind can offer only limited aid in increasing the harvested energy, however further

Moreover, PTFE the most common material, characterised by combination of low friction coefficient, high stability, low dielectric constant, high mechanical strength and superb plasticity, can be easily used to construct all-weather solar cells.

Another potential technology is photoelectron storage, which assumes that a portion of light of specific wavelength could be absorbed my LLP material to be later irradiated back with a wavelength adequate for fotoactive material to generate current. On the other hand, If we consider LLPs materials as additional component to conventional solar cells gives enormous possibility to design and prepare customized material suitable of specific combination of photoactive materials used in photovoltaic device. Moreover, the variety of LPPs synthesis methods in solid state allows use known and cheap technology to produce those materials. The biggest challenge of use of LLPs is their synchronization with the absorption range of photoactive dye: it need to collect only those section of light spectra “wasted” and to transform into specific light band (afterglow) that can be used during dark hours by PV device. In the case of most commonly used photoactive materials, the absorption band of active layer ranges 350-820 nm, so the best LLP material would be Lu2O3 activated with Tb3+ and co doped Ca2+, Sr2+ excited at 254 nm providing 20-30 hours of afterglow at 543 nm [30]. These parameters fits perfectly materials such as P3HT (absorption maximum at approx. 520 nm) or PCDTBT absorption maximum at approx. 560 nm) used for organic PV construction.

Reviewer 2 Report

Manuscript submitted represents a general and poor review, but no novelty is flagged. Moreover, the figures and tables have very low quality. I do not see any advantage to publish this proposal in Energies Journal. 

Author Response

  1. Manuscript submitted represents a general and poor review, but no novelty is flagged. Moreover, the figures and tables have very low quality. I do not see any advantage to publish this proposal in Energies Journal. 

Ad. 1. I am grateful for the effort in reviewing the manuscript. In order to stressed better the advantages of this review some changes have been made in the manuscript.

Reviewer 3 Report

The article is very interesting and well developed.

In my opinion, it can be improved to facilitate its reading and understanding.

Abbreviation section, review and include missing ones, like PCDTBT (page 13, second paragraph), etc.

Tables, consider to unify the format of them, 6 tables with 3 different format. in my opinion the author must include another ones to summarize the information presented as example in page 8, 10, 15, etc). There are a lot of information in the paragraph whose could be presented more clearly. 

Graphs, most of them with high quality but the size makes reading and interpretation difficult, specially figures 10, 12 and 14 compared with 11 and 13. In the graphs, must be reviewed and included the legend box, facilitates the interpretation rather than placing the legend on the graph itself (as example, fig 9a, fig 14 and fig 16 from c to f).

Citations, need to review. missing citations or cannot be clearly identified some of them, like Wong et al (page 8, last paragraph) or Sun et al (page 14 third paragraph). consider to review again, the newest reference comes from 2018 except one [4] and not directly related with de issue of the article.

Author Response

  1. Abbreviation section, review and include missing ones, like PCDTBT (page 13, second paragraph), etc.

Ad.1. I am grateful for the comment. As suggested I adjusted the abbreviation section as suggested. It is now placed at the end of the manuscript in updated version.

  1. Tables, consider to unify the format of them, 6 tables with 3 different format. in my opinion the author must include another ones to summarize the information presented as example in page 8, 10, 15, etc). There are a lot of information in the paragraph whose could be presented more clearly. 

Ad.2. I am thankful for the comment. I unify the format of all tables according to Energies formatting.

  1. Graphs, most of them with high quality but the size makes reading and interpretation difficult, specially figures 10, 12 and 14 compared with 11 and 13. In the graphs, must be reviewed and included the legend box, facilitates the interpretation rather than placing the legend on the graph itself (as example, fig 9a, fig 14 and fig 16 from c to f).

Ad.3. I am grateful for the comment. I change the images with better resolution.

  1. Citations, need to review. missing citations or cannot be clearly identified some of them, like Wong et al (page 8, last paragraph) or Sun et al (page 14 third paragraph). consider to review again, the newest reference comes from 2018 except one [4] and not directly related with de issue of the article.

Ad.4. I am grateful for the comment. I did revise the citation and added missing references in the text. Regarding the year of the cited work the most valuable work was published before 2018 regarding the subject of this review.

Reviewer 4 Report

The topic is interesting and it is adapt to this journal. The collaboration among several faculties is useful and I think that there is a great work behind the presentation of this work. The manuscript and the research in it are well structured. However, in my opinion, the paper is sometimes difficult to follow and more information is required on some issues. My comments:

-Clarify better the innovation of this work in the abstract and in the main text.

-Please, avoid lumped references. What is the contribution of each reference?

-It would be better to use Energies' "Microsoft Word template" document for the manuscript:

https://www.mdpi.com/journal/energies/instructions

-The abbreviations section should be prepared after the "Conflicts of Interest" section, see example: https://www.mdpi.com/1996-1073/14/14/4074/htm

-Extend the conclusion with more general usability. What are the benefits of the results in a global context? Please explain this better in the manuscript.

Author Response

  1. My comments:

-Clarify better the innovation of this work in the abstract and in the main text.

-Please, avoid lumped references. What is the contribution of each reference?

-It would be better to use Energies' "Microsoft Word template" document for the manuscript: https://www.mdpi.com/journal/energies/instructions

-The abbreviations section should be prepared after the "Conflicts of Interest" section, see example: https://www.mdpi.com/1996-1073/14/14/4074/htm

-Extend the conclusion with more general usability. What are the benefits of the results in a global context? Please explain this better in the manuscript.

Ad.1. I am grateful for all comments and suggestions. Revised version contains all above mentions weak points which are marked in blue ink.

Round 2

Reviewer 2 Report

I appreciate the efforts done by the author to improve the manuscript. However, I am sorry to inform that in my point of view, the document is not ready enough to be published in this journal.